# Bdelloid rotifers deploy horizontally acquired biosynthetic genes against a fungal pathogen

Reuben W. Nowell [1,2,3,4], Fernando Rodriguez [5], Bette J. Hecox-Lea [5], David B. Mark Welch [5], Irina R. Arkhipova [5], Timothy G. Barraclough[1,2] & Christopher G. Wilson [1,2] ✉

Coevolutionary antagonism generates relentless selection that can favour genetic exchange, including transfer of antibiotic synthesis and resistance genes among bacteria, and sexual recombination of disease resistance alleles in eukaryotes. We report an unusual link between biological conflict and DNA transfer in bdelloid rotifers, microscopic animals whose genomes show elevated levels of horizontal gene transfer from non-metazoan taxa. When rotifers were challenged with a fungal pathogen, horizontally acquired genes were over twice as likely to be upregulated as other genes − a stronger enrichment than observed for abiotic stressors. Among hundreds of upregulated genes, the most markedly overrepresented were clusters resembling bacterial polyketide and nonribosomal peptide synthetases that produce antibiotics. Upregulation of these clusters in a pathogen-resistant rotifer species was nearly ten times stronger than in a susceptible species. By acquiring, domesticating, and expressing non-metazoan biosynthetic pathways, bdelloids may have evolved to resist natural enemies using antimicrobial mechanisms absent from other animals.

Antagonistic interactions among species are strong, ubiquitous and relentless sources of selection in natural populations[1–3]. Examples include arms-races between pathogen virulence factors and host immune systems, and coevolution between antimicrobial compounds or toxins and pathways to resist them. These dynamics are central to various global challenges, including emerging infectious diseases, management of crop pathogens, and antimicrobial resistance. According to theory, selection arising from antagonistic coevolution can favour adaptations to shuffle existing combinations of genes or to acquire genes bringing new functions. These processes help explain the especially rapid and intense adaptive evolution[4] seen at loci encoding the molecular mediators of conflict[5].

Different domains of life typically address the challenge of ongoing genetic mixing in distinct ways. In bacteria and archaea, horizontal gene transfer (HGT) occurs by various mechanisms[6,7] and is well known as a route for the spread of antimicrobial resistance[8], as well as genes encoding the production of antibiotics and other molecular weapons[9]. For example, nonribosomal peptide synthetases (NRPS) and polyketide synthetases (PKS) are large, multimodular enzymes that catalyse assembly of a vast array of natural products, including toxins, immunosuppressants and antimicrobial compounds[10]. These can be encoded as biosynthetic gene clusters on plasmids[11] as well as chromosomes, and their mobility and 'assembly line' structure facilitate diversification to produce novel secondary metabolites via recombination of modules within and between genomes[10]. The natural

[1]Department of Biology, University of Oxford, 11a Mansfield Road, Oxford OX1 3SZ, UK. [2]Department of Life Sciences, Imperial College London; Silwood Park Campus, Buckhurst Road, Ascot, Berkshire SL5 7PY, UK. [3]Institute of Ecology and Evolution, University of Edinburgh; Ashworth Laboratories, Charlotte Auerbach Road, Edinburgh EH9 3FL, UK. [4]Biological and Environmental Sciences, University of Stirling, Stirling FK9 4LA, UK. [5]Josephine Bay Paul Center for Comparative Molecular Biology and Evolution, Marine Biological Laboratory, Woods Hole, MA, USA. ✉e-mail: chris.wilson@biology.ox.ac.uk

reservoir of mobile genetic diversity for antibiotic synthesis and resistance is thought to reflect a history of coevolution between the producers and targets of antimicrobial compounds[12].

In contrast, among eukaryotes, the most important mechanism of genetic exchange is meiotic sex, by which whole genomes are shuffled every generation though recombination, segregation and outcrossing. Although meiotic shuffling has different effects to HGT[13], sex too has been linked to biotic conflict because it can speed up host adaptation against pathogens by generating new combinations of resistance alleles[14,15]. When coevolving pathogens are common and virulent[16], the theoretical benefits of genetic exchange are so substantial that they may outweigh the inherent costs[17] of sexual reproduction compared with parthenogenesis[18–20]. Antagonistic coevolution may therefore help explain why obligately asexual plant and animal lineages are typically rare and short-lived despite major advantages[21,22]. This so-called 'Red Queen Hypothesis'[3] (RQH) draws support from associations between recombination and immunity, from host-pathogen dynamics in mixed sexual and asexual populations[23,24], and from the susceptibility of asexually propagated lineages to pathogens[25].

Here, we investigate the links between genetic transfer and biotic conflict in a group of animals that challenge typical distinctions between domains described above. Bdelloid rotifers are a class of microscopic, filter-feeding invertebrates that live in freshwater and limnoterrestrial habitats worldwide. Reproduction is only known by parthenogenetic eggs, produced by a modified, nonreductional meiosis[26]. Neither males nor sperm have been reported despite centuries of microscopic observation and the description of hundreds of species[27], leading to the hypothesis that the class Bdelloidea has diversified for tens of millions of years in the absence of sexual reproduction[28,29]. Genetic evidence to either confirm or refute obligate asexuality in bdelloids has proved complicated[30], and seemingly definitive evidence both for and against this hypothesis[31–35] has been overturned or reinterpreted by later work[36–42]. In contrast, repeated studies have demonstrated that bdelloid genomes encode extraordinarily high proportions of genes acquired horizontally from non-metazoan taxa[30]. Approximately 10% of genes appear to have been captured from bacteria, fungi, plants and other sources, rather than sharing recent common ancestry with metazoan orthologs[33,43,44]. This estimate is an order of magnitude greater than for other animals[45], holds for all bdelloid genomes so far examined, and is consistent across various methods for detecting HGT[37,46–48].

Comparisons among bdelloid species indicate that most HGT events are ancient, with ongoing acquisition rates estimated to be on the order of one gene per 100,000 years[49]. At these rates, the phenomenon would be too slow to equate with the sexual shuffling seen in typical eukaryotes, or the rapid dynamics of bacterial accessory genomes. Nevertheless, HGT has been hypothesised to introduce novel biochemical functions that help bdelloids adapt to environmental challenges, as it does in bacteria. Acquired genes are expressed and incorporated into metabolic pathways[44,46], some of which are not shared by other metazoans[50]. Putative functions identified to date include desiccation tolerance, nutrient exploitation and repair of DNA damage[44,46,51,52]. However, the deep associations between genetic transfer and coevolution raise the hypothesis that HGT may help bdelloids deal with biotic antagonism; for instance by acquiring genes with pathogen resistance functions. Isolated examples of horizontally acquired genes contributing to immunity are known from invertebrates[53–55], but the massive scale of HGT in bdelloids and the prevalence of asexual reproduction might especially favour the co-option of unusual pathways to resist microbial enemies, with parallels to the role of HGT in bacterial conflict[56,57]. If so, this could compensate in part for the challenge that an asexual lineage theoretically faces from pathogens.

We investigated this hypothesis by testing whether horizontally acquired genes show a potential enrichment in the response of bdelloid rotifers to infection. Like all animals, bdelloid rotifers are exploited by a range of natural enemies, including over 60 species of virulent fungal and oomycete pathogens[58]. These can exterminate cultured populations in a few weeks[59,60], and significantly depress the abundance of hosts in natural habitats[61]. However, almost nothing is known about variation in susceptibility among rotifers, how this compares with observations in other invertebrate pathosystems[62–67], or how the underlying mechanisms evolve and remain effective if sex is rare or absent.

We used RNA-seq to identify genes that are differentially expressed when bdelloid rotifers are attacked by a natural fungal pathogen in the genus *Rotiferophthora*[68] (*Clavicipitaceae*, *Hypocreales*), which preys specifically upon them (Fig. 1a). We assessed variation by comparing two host species, *Adineta ricciae* and *A. vaga*, which differ by a factor of four in resistance to this pathogen (Fig. 1b). We compared the scale and speed of the transcriptomic response and asked whether horizontally transferred genes were especially likely to be differentially expressed compared to regular metazoan genes in each species. We compared our results with RNA-seq data obtained when bdelloids were exposed to desiccation[51], to test whether genes of non-metazoan origin contribute differently when responding to a biotic as opposed to an abiotic stressor. Functional enrichment analysis identified the most strongly upregulated classes of horizontally transferred genes, and revealed groups whose expression profiles and genomic representations differed between the resistant and susceptible rotifer species.

## Results

### Susceptibility to a fungal pathogen differs markedly between two *Adineta* species

We inoculated clonal populations of the bdelloid rotifers *A. ricciae* and *A. vaga* with spores of the fungal pathogen *Rotiferophthora globospora*[68] and monitored them over 3–4 days. In both species, >95% of animals ingested spores and contracted within 60 minutes of exposure (Fig. 1a). In a successful infection, the ingested conidium germinates within approximately 7 hours of exposure. By 24 hours it has differentiated into assimilative hyphae that begin to invade the host's oesophageal tissue[68], eventually killing and emerging from the host after approximately 48–72 h (Supplementary Fig. 1). The proportion of animals killed by fungal infection at 72 h differed markedly between the species: 18% for *A. ricciae* and 71% for *A. vaga* (Fig. 1b, relative risk = 3.74, 95% CI: 2.8–5.0, $z = 8.758$, $P < 0.001$). Given that these species are morphologically[37,69] similar, were reared and exposed under standardised conditions, the difference in mortality seems to indicate variation in the physiological response and genetic resistance of each host to the fungus.

### Rapid and large-scale transcriptional response to pathogen attack in both species

To investigate the genetic basis of the response to pathogens, we harvested total RNA from populations of each species in triplicate at 7 and 24 hours after exposure to live or UV-inactivated fungal spores, and used gene models predicted from the reference genomes for *A. vaga*[33] ('Av13') and *A. ricciae*[37] ('Ar18') to quantify gene expression (Supplementary Data 1). These timepoints (T7 and T24, respectively) cover the key early stages of the fungal attack (Supplementary Fig. 1). Principal component analysis (PCA) of overall gene expression showed consistent clustering of replicate samples within and between species (Fig. 1c). At the broadest scale, this indicates a shared transcriptional response to *R. globospora* infection that is particularly similar at the earlier timepoint (Fig. 1c; Supplementary Data 2). To investigate the genes involved in this response, we defined significant differential expression (DE) as an absolute fold-change > 4 and a False Discovery Rate-adjusted *P*-value < 0.001 (Supplementary Data 1). Even by these stringent criteria, a rapid and pronounced transcriptional

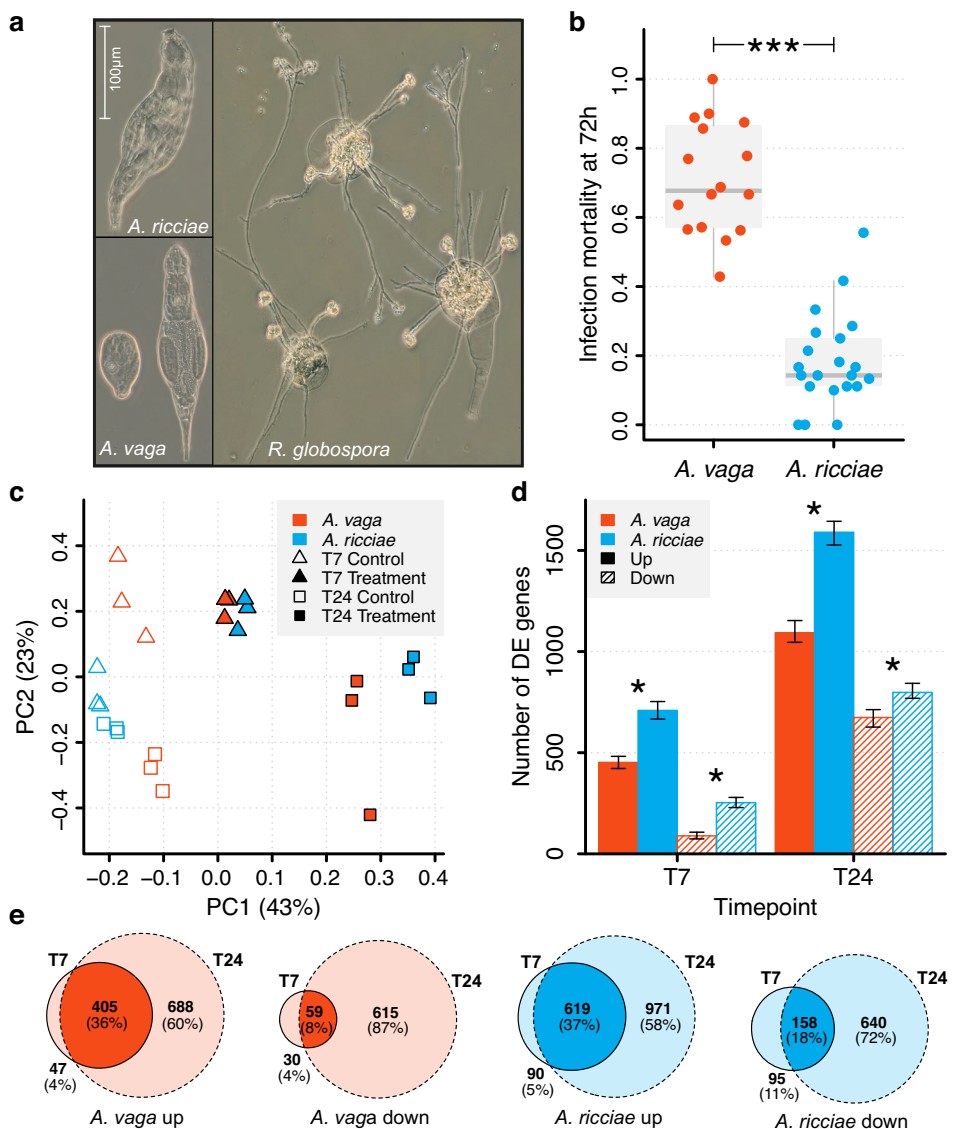

**Fig. 1 | Response of bdelloid rotifers to inoculation with the fungal pathogen *R. globospora*. a** Active *A. ricciae*; active and contracted *A. vaga*; composite of three *A. ricciae* corpses with fully developed *R. globospora* infections, differentiating into irregular resting spores and long conidiophores bearing spherical infectious conidia. **b** Proportion of rotifers killed by infection 72 hours after exposure to *R. globospora*. Points indicate replicate laboratory populations of *A. vaga* ($n = 16$ populations, 189 animals) and *A. ricciae* ($n = 21$ populations, 216 animals) exposed to 1000 conidia. Asterisks indicate a highly significant difference in infection mortality (relative risk test, RR = 3.74, 95% CI: 2.8–5.0, $z = 8.76$, $P = 1.4e{-}25$). Boxplots show median and interquartile range (IQR), and whiskers extend to the farthest datapoint from the median that remains within 1.5*IQR of Q1 and Q3 respectively. **c** Principal component analysis of overall gene expression across *A. vaga* and *A. ricciae* orthologous genes. The proportions of total variation accounted for by primary (PC1) and secondary (PC2) components are indicated in parentheses. Clustering of treatment groups across species indicates a strongly shared response in gene expression at T7 that diverges at T24, as *A. ricciae* moves further and more consistently along the PC1 axis than *A. vaga*. **d** Dynamics of gene up- and downregulation relative to control populations inoculated with UV-inactivated spores. Error bars around the observed number of genes in each category show 95% confidence intervals estimated from $n = 100$ bootstrap replicates (i.e., sampling across all genes with replacement 100 times, each time recalculating the number of shared genes in each category). Asterisks refer to significant differences in the proportion of the species' respective genesets that are DE (Bonferroni-corrected Chi-square tests, d.f. = 1; $n = 58,423$ for *A. ricciae*, 66,273 for *A. vaga*; $P = 9.7e{-}22$, $5.4e{-}23$, $4.7e{-}38$ and $5.8e{-}8$ respectively). **e** Extent of gene sharing in differentially expressed subsets across timepoints (within species). Values show the number of significantly DE genes shared by intersecting groups. Source data are provided as a Source Data file.

response was detected in both species, involving hundreds of differentially expressed genes (Fig. 1d). At T7, 541 *A. vaga* genes and 962 *A. ricciae* genes showed significant DE. Most were upregulated (452 in *A. vaga*, ~84%; 709 in *A. ricciae*, ~74%). At T24, the number of DE genes rose to 1767 in *A. vaga* (1093 upregulated, ~62%), and 2388 in *A. ricciae* (1590 upregulated, ~67%). Thus, more genes showed significant DE at T24 than at T7 in both species, but more genes were DE (and especially upregulated) in *A. ricciae* than in *A. vaga*. The larger number of responding genes (Fig. 1d) and the greater magnitude of upregulation in *A. ricciae* (Supplementary Table 1) suggest a broader and stronger

defensive response than *A. vaga*. Both species showed a substantial overlap in the identity of differentially expressed genes across timepoints, particularly for upregulated subsets (Fig. 1e), while the direction and magnitude of DE at T7 correlated significantly with that seen at T24 (Pearson's correlation $R = 0.66$–$0.84$, $P < 2e{-}16$ in all cases; Supplementary Fig. 2). This suggests that many of the transcriptional changes initiated at T7 were ongoing and consistent at T24, even as new genes joined the response.

Dynamics in expression among *A. vaga* and *A. ricciae* orthologs were also examined. Again, the two species showed substantially

overlapping profiles of differentially expressed genes and significant positive correlations between orthologous gene copies both within and between genomes, at both timepoints (Pearson's correlation $R = 0.35$–$0.60$, $P < 2e$–$16$ in all cases; Supplementary Figs. 3 and 4, and Supplementary Table 2), as expected if pathogen response pathways are conserved. Both species, therefore, mounted rapid, extensive, and partially overlapping transcriptional responses to the pathogen, beginning less than 7 hours after exposure.

## Twofold overrepresentation of non-metazoan genes in the response to pathogens

Based on the predicted proteins in each reference genome, the background frequency of high-confidence HGT candidate genes (denoted '$HGT_C$') is ~11.5% for both species[48]. Among the significantly upregulated subset, however, 23–32% were $HGT_C$, depending on species and timepoint (Table 1). This two- to three-fold enrichment for $HGT_C$ was highly significant (binomial GLM, estimate = 1.18, SE = 0.07, $P < 0.0001$; Fig. 2; Supplementary Table 3) in both species at both timepoints. The proportional enrichment was significantly stronger for the early response (estimate = 0.22, SE = 0.08, $P = 0.0048$), though the absolute numbers of upregulated $HGT_C$ genes were higher at T24 (Fig. 2 and Table 1; Supplementary Table 3). We observed a similar enrichment of $HGT_C$ among significantly downregulated genes at both experimental timepoints (estimate = 1.09, SE = 0.12, $P < 0.0001$; Fig. 2 and Table 1; Supplementary Table 3), which again did not differ between the two species. Horizontally acquired genes were therefore markedly overrepresented among the genes that are differentially expressed in the pathogen treatments in both species, accounting for about 33% and 25% of the most strongly responding genes at T7 and T24, respectively. Of the total $HGT_C$ complement in *A. ricciae* and *A. vaga*, 4.2% and 2.1% were differentially expressed at T7, rising to 9% and 5.7% respectively at T24.

## Enrichment of horizontally acquired genes is stronger when responding to pathogens than to desiccation

We considered whether the apparent enrichment of $HGT_C$ was a feature of the response to fungal attack specifically, or might result from more generic properties of such genes and their expression. For

example, $HGT_C$ might tend to show differential expression between any given pair of conditions or timepoints if they are overrepresented among effectors at the ends of gene regulatory networks, or if their expression level is less precisely regulated than core metazoan genes[70,71]. Alternatively, $HGT_C$ might be overrepresented among general stress-response genes, rather than those responding to pathogens in particular[44,46,51].

To test these possibilities, we repeated the analyses above using RNA-seq data from a prior study[51] that investigated gene expression when the same cultured strain of *A. vaga* was exposed to desiccation stress. Most bdelloid species can tolerate complete loss of cellular water by entering a state of physiological dormancy known as anhydrobiosis[72,73]. Hecox-Lea and Mark Welch (2018) compared gene expression in hydrated animals to those that were either entering or recovering from anhydrobiosis, and we tested for enrichment of $HGT_C$ in these responses. The proportion of $HGT_C$ among genes significantly upregulated in response to desiccation was substantially lower than in the pathogen experiment (13.8% for 'entering', 14.7% for 'recovering', compared to ~23–32% in the pathogen experiments; Fig. 3 and Table 1; Supplementary Data 1). There was no significant $HGT_C$ enrichment when animals were entering desiccation (estimate = 0.14, SE = 0.12, $P = 0.26$; Fig. 3a and Table 1; Supplementary Table 4), but significant enrichment was seen during recovery (estimate = 0.24, SE = 0.07, $P = 0.0004$; Fig. 3b and Table 1; Supplementary Table 4), especially for downregulated genes (estimate = 0.64, SE = 0.07, $P < 0.0001$). The markedly lower enrichment of $HGT_C$ is unlikely to reflect a less extensive transcriptional response overall because we detected nearly twice as many differentially expressed genes during recovery from desiccation ($n = 3129$) versus pathogen challenge ($n = 1767$). While datasets from different experiments should be compared with caution, a joint PCA also indicated that changes in overall gene expression were at least as pronounced for the abiotic stressor as for the pathogen, relative to their respective control groups (Supplementary Fig. 11). Nevertheless, the enrichment of $HGT_C$ genes was between two and three times stronger when responding to pathogens compared with desiccation. To test how far the transcriptional responses are specific to each challenge or reflect general stress response pathways, we

**Table 1 | Number of $HGT_C$ in significantly up- and downregulated gene subsets in response to biotic (pathogen exposure) and abiotic (desiccation) stress**

| Test[a] | Species | Contrast | DE subset[b] | $HGT_C$[c] in: | | Odds ratio (95% CI) | P-value[d] |
|---|---|---|---|---|---|---|---|
| | | | | DE0 | DE1 | | |
| Pathogen | *A. ricciae* | T7 | Up | 6041 / 57461 (11%) | 196 / 709 (28%) | 3.3 (2.7–3.8) | 2.1e–36 |
| | | | Down | | 68 / 253 (27%) | 3.1 (2.3–4.2) | 3.1e–13 |
| | | T24 | Up | 5738 / 56035 (10%) | 372 / 1590 (23%) | 2.7 (2.4–3.0) | 5.8e–50 |
| | | | Down | | 195 / 798 (24%) | 2.8 (2.4–3.3) | 4.9e–30 |
| | *A. vaga* | T7 | Up | 7918 / 65732 (12%) | 145 / 452 (32%) | 3.4 (2.8–4.2) | 6.1e–29 |
| | | | Down | | 26 / 89 (29%) | 3.0 (1.8–4.8) | 1.2e–5 |
| | | T24 | Up | 7629 / 64506 (12%) | 285 / 1093 (26%) | 2.6 (2.3–3.0) | 3.2e–37 |
| | | | Down | | 175 / 674 (26%) | 2.6 (2.1–3.0) | 1.8e–23 |
| Desiccation | *A. vaga* | Entering | Up | 7975 / 65354 (12%) | 77 / 558 (14%) | 1.2 (0.9–1.5) | 0.24 |
| | | | Down | | 37 / 362 (10%) | 0.8 (0.6–1.2) | 0.29 |
| | | Recovering | Up | 7553 / 63144 (12%) | 266 / 1807 (14%) | 1.3 (1.1–1.5) | 5.6e–4 |
| | | | Down | | 270 / 1322 (20%) | 1.9 (1.6–2.2) | 5.9e–18 |

[a]Test: 'Pathogen', treatment groups represent animals exposed to live pathogen spores, versus controls exposed to inactivated spores, at 7 h (T7) and 24 h (T24) post-inoculation; 'Desiccation', treatment groups represent animals either taken from wet dishes (hydrated controls), or equivalent dishes left without lids for 2-4 days until only a thin water film remained (entering into desiccation), or animals left for a further 4 days to enter full desiccation for 7 days, then harvested 1 h after rehydration[51].
[b]DE subset: 'Up', genes that are significantly upregulated in the treatment group; 'Down', genes that are significantly downregulated in the treatment group. Significance is defined as genes with absolute fold change in expression > 4 and FDR < 1e–3.
[c]The number of $HGT_C$ in different subsets as follows: 'DE0', genes with no significant change in expression (either up or down); 'DE1', genes significantly up- or downregulated, depending on the defined subset. Fraction denominators show the total number of genes in each subset.
[d]P-value for Fisher's exact test (two sided) for an association between classifications; null hypothesis: true odds ratio is equal to 1.

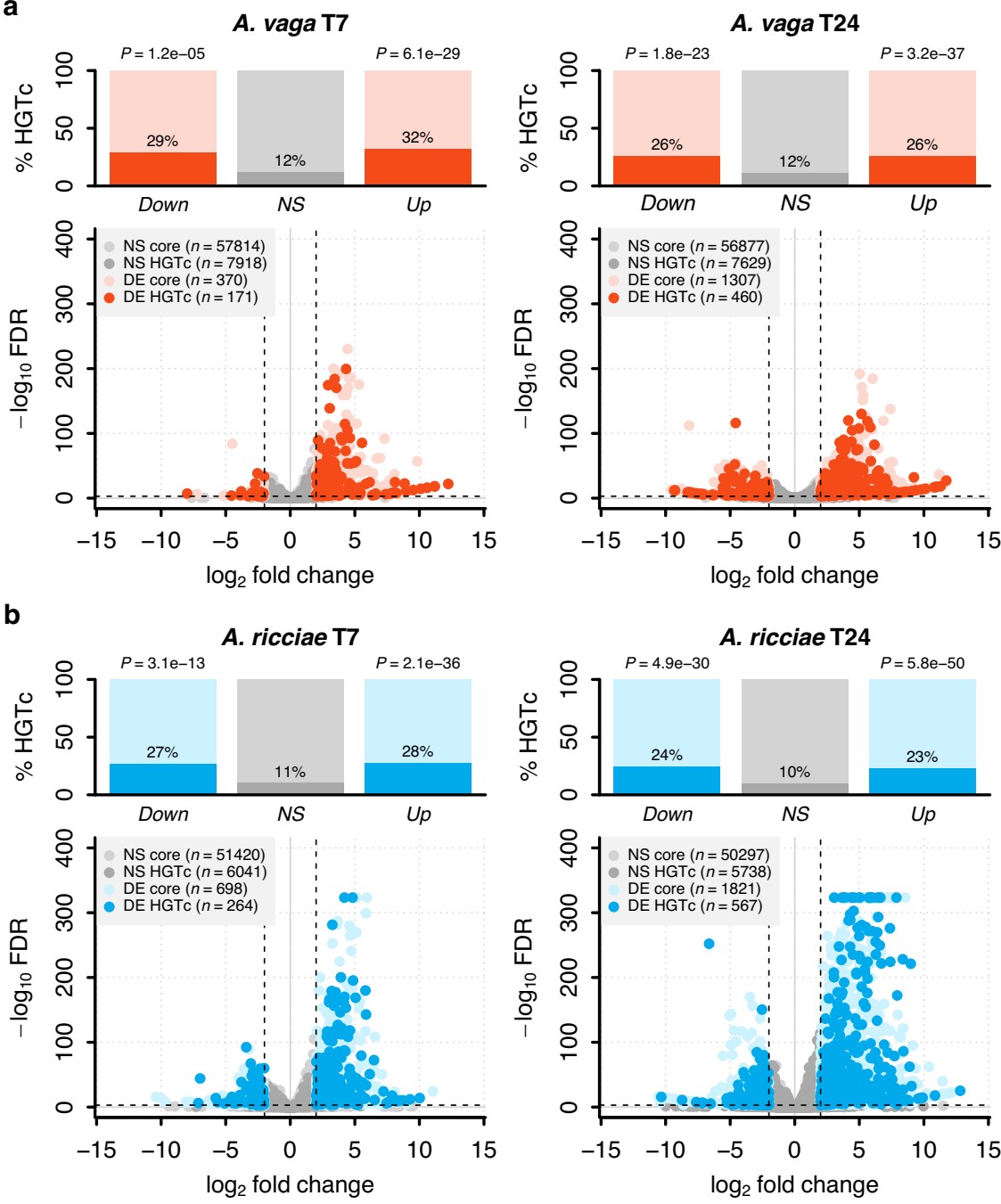

**Fig. 2 | Gene expression in response to pathogen exposure in the bdelloid rotifers *A. vaga* and *A. ricciae*.** Points represent individual genes plotted by $\log_2$ fold-change in expression level on the *X*-axis and significance ($-\log_{10}$ FDR) on the *Y*-axis, shown for **a** *A. vaga* and **b** *A. ricciae* at timepoints 7- and 24-hours post-inoculation. Positive *X*-axis values indicate genes upregulated in response to live pathogen challenge. Genes with significant expression changes (defined as absolute fold-change > 4 and FDR < 1e−3, dashed lines) are shown in colour, with HGT$_C$ indicated by darker shading. Genes with non-significant expression changes are shown in grey (> 95% of genes are non-significant at these thresholds). At T7, the relative magnitude of DE among upregulated genes did not differ significantly between species (estimate = 0.076, SE = 0.053, $t = 1.44$, $P = 0.15$). However, at T24

the relative magnitude of upregulation was significantly higher for *A. ricciae* than *A. vaga* (three-way interaction of time, species and DE set: estimate = 0.23, SE = 0.058, $t = 4.02$, $P = 5.8\mathrm{e}{-5}$). No significant differences were detected between species in magnitude of downregulation (Supplementary Table 1). 'NS core' = no significant change in expression and not HGT$_C$; 'NS HGT$_C$' = no significant change in expression and is HGT$_C$; 'DE core' = significant change in expression (up or down) and not HGT$_C$; 'DE HGT$_C$' = significant change in expression (up or down) and is HGT$_C$; values indicate the number of genes in each category. Bar plots show the proportion (%) of HGT$_C$ per DE subset. *P*-values refer to tests of non-association between HGT$_C$ and the corresponding DE subset (two-tailed Fisher's exact tests, Table 1).

checked the identities of genes that were differentially expressed in the two experiments. Overall, the proportion of upregulated genes shared between any two treatment conditions was low (mean = -13%; Fig. 3c and Supplementary Table 5), indicating that the large majority

of significantly differentially expressed genes in *A. vaga*, including HGT$_C$, are specific to either the infection or desiccation experiment.

These results also are consistent with evidence from a third experiment studying the response of *A. vaga* to desiccation and

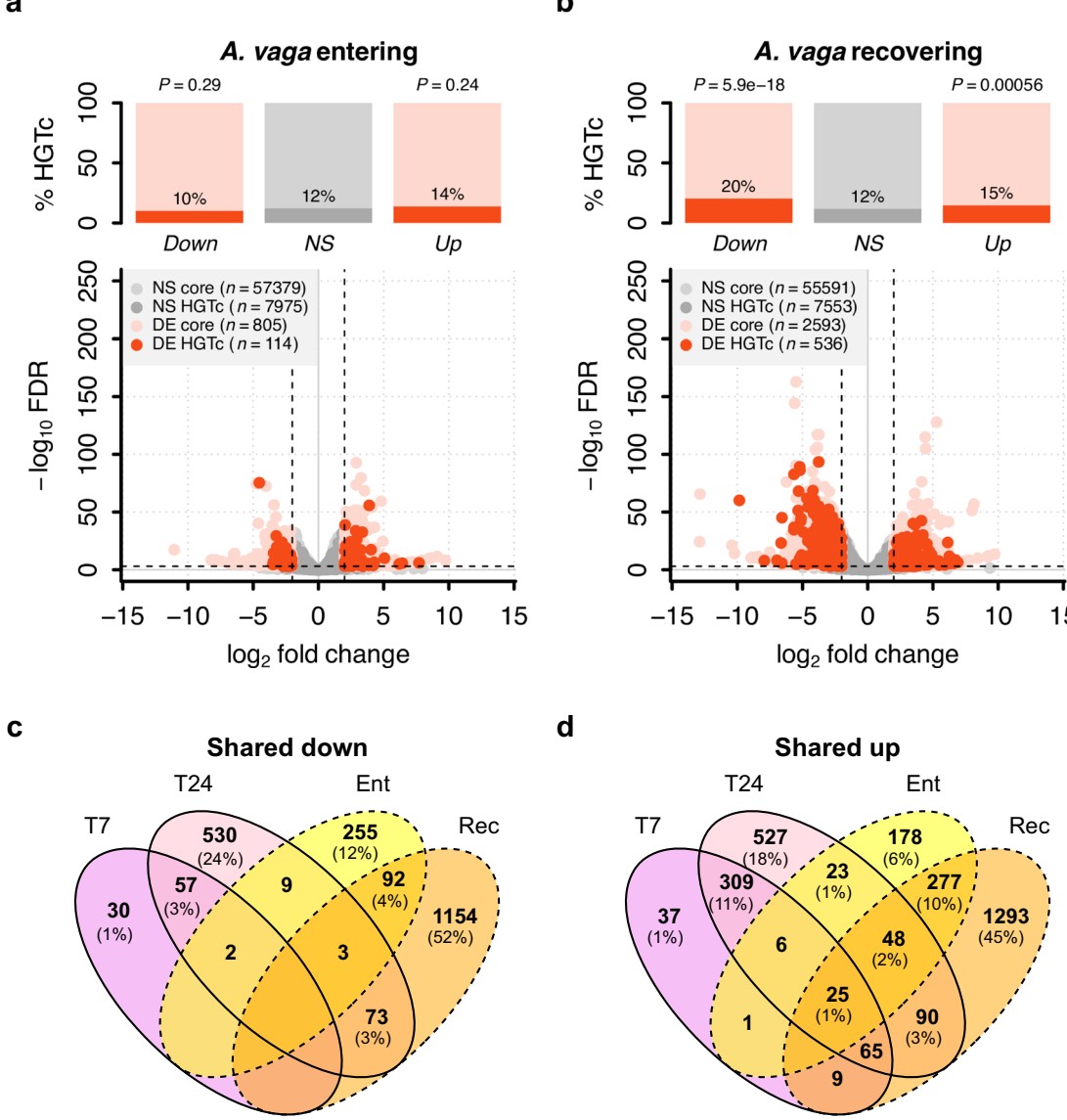

**Fig. 3 | Gene expression in response to desiccation stress in *A. vaga.*** Reanalysis of gene expression data from Hecox-Lea and Mark Welch (2018)[51], showing the proportion of HGT_C in significantly DE gene subsets when animals are **a** entering and **b** recovering from desiccation. Plots arranged as in Fig. 2; enrichment relative to non-DE baselines were assessed as above by two-tailed Fisher's exact tests. Extent of gene sharing in differentially (**c**) upregulated and **d** downregulated subsets between the pathogen and desiccation experiments. Pathogen groups (T7 and T24) are shown in solid outlines, desiccation groups ('Ent' = entering anhydrobiosis and 'Rec' = recovering from desiccation) in dashed outlines. Values in each segment show the number of *A. vaga* genes significantly up- or downregulated for intersecting groups. Segments with no values have no genes shared across that intersection. Overall, the proportion of upregulated genes shared between experiments is low (ca. 10%) compared to within experiments (mean = 52%; see Supplementary Table 5 for further details). Values just for HGT_C showed a similar pattern: of the 285 HGT_C upregulated during recovery from desiccation, only 33 (-12%, T7) and 63 (-24%, T24) were also upregulated in response to pathogens. Even fewer genes are shared among downregulated subsets.

ionising radiation[74]. The proportion of HGT_C among upregulated genes in the desiccation response (14.3%) was nearly identical to that reported above (14.7%). The radiation response overlapped only partially with desiccation, and 10.3% of upregulated genes were reported as HGT_C, which was not a significant enrichment. Genes upregulated in response to pathogens in our experiment thus showed significantly stronger enrichment for HGT_C than has been reported to date for two abiotic stressors.

### The susceptible and resistant species differ strikingly in expression of HGT_C with predicted biosynthetic functions

To investigate putative functions of the upregulated genes, we performed functional enrichment analyses for Gene Ontology (GO)

terms[75] overrepresented among upregulated HGT_C, relative to all genes (Supplementary Data 3). We focused first on *A. ricciae* at T24, taking this as a mature expression profile that ought to include genes mediating resistance, given that over 70% of these hosts eventually survived. Applying stringent statistical criteria (FDR < 0.001), we identified 29 significantly enriched GO terms among upregulated HGT_C (summarised in Fig. 4a). The most highly overrepresented term was 'phosphopantetheine binding' (GO:0031177). Phosphopantetheine is a key cofactor in the activity of non-ribosomal peptide, polyketide and hybrid synthetases (NRP/PKS), acting as a 'swinging arm' to bring activated fatty acid or amino acid groups into contact with sequential catalytic centres during biosynthesis[76,77]. Three more of the top five most strongly enriched

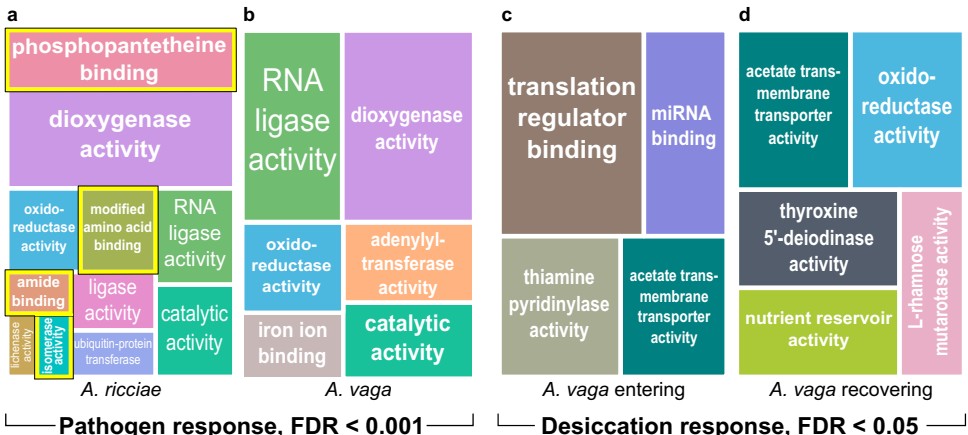

**Fig. 4 | Enriched Gene Ontology (GO) terms for upregulated HGT$_C$ at T24.**
**a** Significantly enriched GO terms relating to the molecular function of HGT$_C$ upregulated in *A. ricciae* in response to the pathogen (FDR < 0.001) at T24. The area of the rectangle corresponds to the relative magnitude of enrichment of each term; related sub-terms are grouped under a single colour. Terms associated with NRP/PKS functions are highlighted with a yellow border. 'Catalytic activity' and 'oxidoreductase activity' are associated with NRP/PKS functions too, but these are high-level generic terms. **b** Equivalent plot for *A. vaga*. **c** As a control, we applied the same functional enrichment analysis (but with a permissive FDR < 0.05) to HGT$_C$ that were differentially expressed by *A. vaga* while entering desiccation (Supplementary Data 5). **d** Equivalent plot showing GO terms for upregulated HGT$_C$ (FDR < 0.05) during recovery from desiccation. If enriched gene categories reflect a generalised stress response rather than putative adaptations to a fungal attack, or arise from biases in GO or HGT$_C$ annotation, we would expect some of the same terms to emerge. However, there was no enrichment in either case for GO terms

relating to NRP/PKS. Instead, NRP/PKS-associated terms were significantly enriched among genes downregulated by *A. vaga* when entering or recovering from desiccation (e.g. 'phosphopantetheine binding', FDR = 1.45e−5; 'antibiotic biosynthetic process', FDR = 0.0029; Supplementary Data 5). In response to desiccation, therefore, rotifers seem more likely to downregulate NRP/PKS genes that had been constitutively expressed in hydrated control animals, rather than upregulating those that were not previously active. RNA ligase and glucan-binding (lichenase) functions were not enriched among upregulated genes in the desiccation conditions either, even with a relaxed threshold (FDR < 0.1). When the shared 'oxidoreductase activity' term was disaggregated into more specific sub-terms, there was no overlap between stressors (Supplementary Data 6). In general, almost no overlap was detected in terms of molecular function or biological process between the pathogen and desiccation responses, either for the HGT$_C$ shown here or for all genes considered together (6/157 shared molecular function terms, < 4%, Supplementary Data 6).

GO terms were similarly associated with NRP/PKS functions or processes. Phosphopantetheine binding remained the most strongly enriched GO term among all upregulated genes (i.e., before filtering for HGT$_C$), indicating that enrichment is not simply because genes with this term are especially likely to be classed as HGT$_C$ (Supplementary Data 4).

The more susceptible species shows a markedly different pattern. Applying the same analysis to *A. vaga* at T24, we found no significant enrichment among upregulated HGT$_C$ for any of eight NRP/PKS-associated GO terms that were enriched in *A. ricciae* (Fig. 4b; Supplementary Data 3), even with relaxed statistical stringency (FDR < 0.05). This was especially notable because 78% of the enriched (FDR < 0.001) GO terms in *A. vaga* were otherwise shared with *A. ricciae* (14/18 terms), rising to 100% overlap at FDR < 0.05. The divergent pattern with respect to NRP/PKS biosynthesis-associated terms, therefore, appears to be the most prominent difference between the resistant and the susceptible species in the predicted functional profiles of upregulated HGT$_C$.

The detailed output of the GO analysis indicated that these results were driven by 27 predicted proteins in *A. ricciae*, each associated with two or more NRP/PKS-related GO terms (inventoried in Supplementary Data 7). An initial BLAST search of these upregulated HGT$_C$ products against the UniProt/Swiss-Prot curated protein database returned top matches to NRP/PKS-like proteins encoded by bacteria and fungi (Supplementary Data 7). Bacterial matches were all NRPS proteins that catalyse the production of antimicrobial compounds, including bacitracin, surfactin, mycosubtilin, tyrocidines and gramicidins− products which show broad-spectrum activity including against fungal hyphae[78–80] and spores[81]. This result raises the hypothesis that *A. ricciae* resists fungal pathogens in part by synthesising antifungal secondary metabolites using highly modified nonribosomal peptide synthetases originally acquired from bacteria.

**Both species encode multiple divergent NRP/PKS clusters, with stronger differential expression by the more resistant species**
To fully survey the *A. ricciae* and *A. vaga* genomes for putative NRP/PKS genes, we searched the predicted proteomes for matches to three key canonical NRP/PKS-related domains (see Methods for details). This returned positive matches in both species: 60 in *A. vaga*, 36 in *A. ricciae* (Supplementary Data 7). The same screen returned no matches to the proteomes of the monogonont rotifers *Brachionus plicatilis*[82] and *B. calyciflorus*[38], nor the acanthocephalan *Pomphorhynchus laevis*[83]. Alignment of the bdelloid sequences to the more comprehensive UniProt/Uniref90 protein database showed hits to a large variety of NRP/PKS-like proteins, overwhelmingly from bacteria, but with a few secondary hits to fungi and protists (Supplementary Data 8). There were no significant hits to other metazoans, suggesting the bdelloid copies are not similar to other known cases of rare NRP/PKS-like genes in animals[84,85]. Phylogenetic analysis of the canonical NRP/PKS condensation domain supported this hypothesis, showing that the majority of bdelloid copies form a large and diverse monophyletic clade distinct from other representatives (Fig. 5a). We observed substantial diversity in domain organisation among bdelloid copies (Fig. 5b), although interpreting these sequences is challenging because NRP/PKS are large multimodular proteins that can be encoded as single genes or clusters. They often contain duplications, recombination or fusion of copies or modules[86] that are not easily resolved in short-read assemblies such as Av13 and Ar18. To check our 'three-domain' NRP/PKS CDS inventory, we used a combination of manual and automated methods to locate putative biosynthetic gene clusters in a recently available chromosome-scale assembly for *A. vaga* (Av20), independent of our other annotation pipelines. We identified approximately 40 clusters (Fig. 5c; Supplementary Fig. 12 and Supplementary Data 9). Like many HGT$_C$, these are overrepresented in highly dynamic subtelomeric regions that are hard to assemble

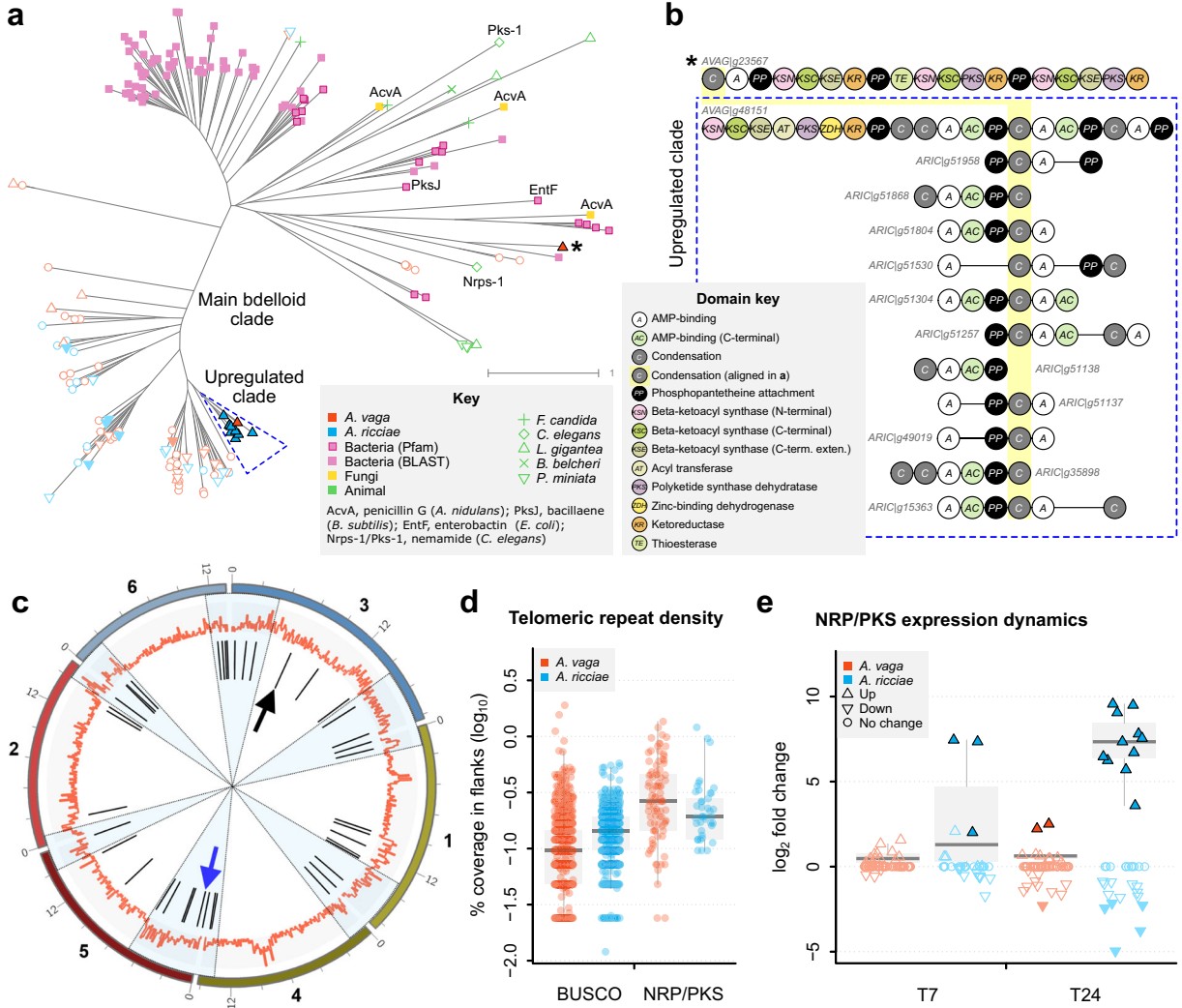

**Fig. 5 | Diversity, structure, genomic location, and expression dynamics of putative NRP/PKS encoded by bdelloid rotifers. a** Phylogeny of NRP/PKS coding sequences (CDS) based on aligning the condensation domain to selected sequences from other kingdoms. Components of selected bacterial and fungal antimicrobial biosynthesis pathways are named for reference, together with Nrps-1 and Pks-1, which synthesise neuron-associated molecules in *Caenorhabditis elegans*. Other examples of animal NRP/PKS include a springtail (*Folsomia candida*), mollusc (*Lottia gigantea*), lancelet (*Brachiostoma belcheri*) and sea star (*Patiria miniata*). Shading and symbols for bdelloid copies correspond to the expression dynamics at T24 (panel e). The 'upregulated clade' in blue dashed lines comprises 11 CDS from *A. ricciae* and 1 from *A. vaga*. The other upregulated *A. vaga* CDS (asterisk) clusters with bacterial homologues and may be a recent acquisition (its best match is pksN1 from *Corallococcus corralloides*; Supplementary Data 8). **b** Domain arrangement for significantly upregulated PKS-NRPS hybrid clusters. Multiple partially-assembled *A. ricciae* CDS are aligned to the more completely assembled, putatively orthologous

PKS-NRPS cluster in *A. vaga* (blue dashed box). The condensation domain used for the phylogeny is highlighted in yellow. **c** Locations of ca. 40 putative biosynthetic gene clusters (black lines, inner circle) in a haploid, chromosome-scale *A. vaga* assembly. Blue-shaded sectors show subtelomeric regions; orange track shows density of HGT$_C$. Arrows demark clusters corresponding to AVAG|g23567 (black) and AVAG|g48151 (blue). **d** Density of a telomeric repeat motif in 25 kb up- and downstream flanking regions surrounding core eukaryotic (BUSCO) genes and putative NRP/PKS ($n$ = 1120, 1138, 97 and 45 for *A. vaga* and *A. ricciae* respectively). Boxplots show median and interquartile range (IQR); whiskers extend to the farthest datapoint from the median that remains within 1.5*IQR of Q1 and Q3 respectively. **e** Expression dynamics for all putative NRP/PKS CDS in *A. vaga* ($n$ = 60) and *A. ricciae* ($n$ = 36) at T7 and T24. Boxplot components are as in d, but show distributions exclusively for upregulated NRP/PKS CDS (significant or not); filled symbols indicate significant DE (absolute fold-change > 4, FDR < 1e−3). Source data are provided as a Source Data file.

(the same holds for *A. ricciae*, Fig. 5d; Supplementary Fig. 13 and Supplementary Table 6). Nevertheless, 59 of 60 (98%) of our NRP/PKS gene models from Av13 mapped to independently identified biosynthetic clusters in Av20, and over 94% of Av20 clusters with a canonical condensation domain were represented at least partially in our putative NRP/PKS gene set. This suggests that the 'three domains' prediction set from the Av13 and Ar18 assemblies gives an accurate, if conservative estimate of NRP/PKS diversity.

Expression profiles of putative NRP/PKS CDS show that the more resistant *A. ricciae* upregulates three at T7, rising to 11 at T24, whereas in the more susceptible *A. vaga*, no NRP/PKS gene models are significantly upregulated at T7, rising to two at T24 (Fig. 5e; Supplementary Data 1). Moreover, the fold-change for upregulated NRP/PKS was over an order of magnitude higher for *A. ricciae* than *A. vaga* at T24, driven by this group of 11. These genes were barely expressed under control conditions: of those with measurable expression, the

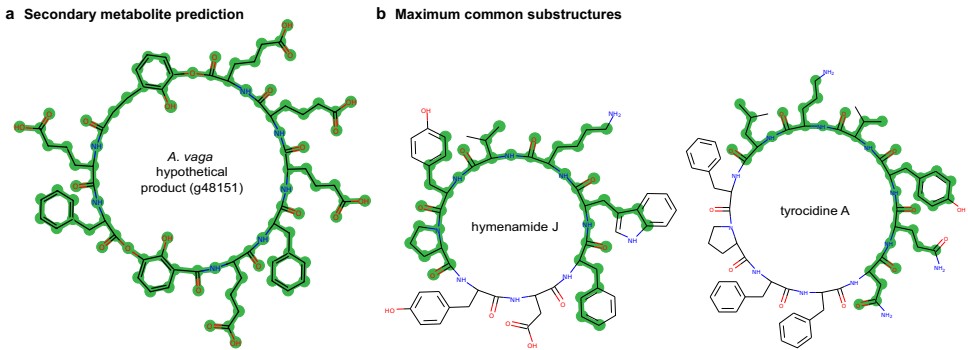

**a** Secondary metabolite prediction    **b** Maximum common substructures

**Fig. 6 | Predicted products of putative NRP/PKS encoded in bdelloid genomes.** **a** Secondary metabolite prediction for the full-length annotation AVAG|g48151 (this was preferred over the partially-assembled annotations depicted in Fig. 5b for the *A. ricciae* ortholog). The PKS-NRPS hybrid cluster corresponding to gene model AVAG|g48151 is predicted by SeMPI 2.0 to synthesise a cyclic heptadepsipeptide. **b** Maximum common substructures estimated by SeMPI 2.0 are shown (green highlighting) for the two named compounds with the highest similarity to the predicted product: hymenamide J, isolated from extracts of the marine sponge *Hymeniacidon* sp. and its microbiota, and tyrocidine A, a cyclic decapeptide with potent antifungal activity produced by *Bacillus brevis*.

mean normalised read count (TMM = 0.015) was lower than 96.8% of genes, reflecting a rapid switch from almost no transcription to large-scale production.

**The resistant species shows duplication of a shared PKS-NRPS hybrid cluster predicted to produce a cyclic peptide similar to known antibiotics**

The condensation phylogeny and expression profile for the set of predicted NRP/PKS CDS shows a closely related (aLRT > 95%) cluster of 11 CDS from *A. ricciae* and 1 from *A. vaga* that are all significantly upregulated ('Upregulated clade' in Fig. 5a), suggesting duplications in *A. ricciae* of an ortholog that is present in both species. To circumvent assembly issues discussed above, we manually investigated copy number and orthology of this focal cluster using alternative assemblies for both *A. vaga* and *A. ricciae* (see Supplemental Methods for further details). This revealed that *A. vaga* encodes two nearly identical homologous copies of this PKS-NRPS hybrid cluster (represented by AVAG|g48151 in Fig. 5b), whereas *A. ricciae* appears to encode at least six copies of an orthologous PKS-NRPS cluster that shares all the key modules. The clusters in *A. ricciae* are flanked by multiple copies of genes encoding the two major classes of metazoan and bacterial metabolite transporter proteins (ABCB1 and MFS-family), associated with resistance to antibiotics and export of toxic compounds from cells. These, too, are dramatically upregulated on exposure to the pathogen (Supplementary Table 7 and Supplementary Data 10), along with a flanking gene encoding the cytochrome P450 domain, linked to secondary metabolite detoxification in metazoans and biosynthesis in bacteria.

As a final step to explore potential function, we predicted the structure of the secondary metabolite synthesised by this PKS-NRPS hybrid cluster, using the full-length annotation represented by AVAG|g48151. The product predicted by SeMPI v2.0[87] is a cyclic heptadepsipeptide (Fig. 6a) with no highly similar hits (metabolite score >0.7) to known compounds in the screened databases (Supplementary Data 11). However, the named bacterial metabolite with the highest similarity (score 0.64) is tyrocidine A (Fig. 6b), a cyclic decapeptide with broad-spectrum antimicrobial activity that disrupts fungal membranes[78], retards conidial germination and inhibits extension of hyphae[88]. A second match (score 0.61) is to hymenamide J, an octapeptide isolated from the microbiota associated with a marine sponge[89], with cytotoxic properties[90] but to our knowledge unscreened for antimicrobial activity (though a related compound, hymenamide E, shows potent antifungal activity[91]).

**HGT_C with matches to RNA ligases and glucanases are disproportionately upregulated in both hosts**

Whereas expression of NRP/PKS CDS differed between the host species, two other prominent HGT_C-encoded elements of the pathogen response were shared. First, terms related to RNA repair are at or near the top of enrichment analyses for both species and timepoints (e.g. GO:0008452, 'RNA ligase activity'; Fig. 4a, b). Only about 15 genes in either species have this term, but 50–70% of these are consistently, significantly and strongly upregulated in response to the fungus (Supplementary Data 3). This includes the two most strongly upregulated genes at both time points in *A. vaga*. The rotifer-encoded RNA ligase domains (Pfam accession PF013563) cluster phylogenetically with non-metazoan lineages (including bacteria, fungi, viruses, and non-metazoan eukaryotes; Supplementary Fig. 14a–c). The top BLAST matches are to RNA ligases from bacteria or bacteriophages, where they function in tail fibre assembly (Supplementary Data 10).

Second, both species show upregulation of HGT_C with putative glucanase activity at both timepoints (e.g. GO:0042972, 'licheninase activity', active on β–1,4 or β–1,3 glycosidic bonds; 6 genes in *A. ricciae*; 3 in *A. vaga* at T24; Supplementary Fig. 14d and e; Supplementary Data 3). These genes had top BLAST matches to glycoside hydrolase (GH) family enzymes in bacteria, and have possible roles in recognising or attacking fungal cell walls[92]. While present in both species, the more resistant *A. ricciae* seems to encode and express a marginally expanded repertoire of these enzymes. Neither RNA ligase nor licheninase functions were enriched in the responses to desiccation (Fig. 4c, d).

A final category that was consistently enriched against pathogens was the generic term 'catalytic activity' (GO:0003824). Closer examination of individual genes revealed at least 6 with matches to components or regulators of plant immunity, including phenylpropanoid and shikimate biosynthetic pathways. Other genes of interest included dioxygenases and two matches to fungal genes linked to programmed cell death: a metacaspase and a caspase-like protease (Supplementary Data 10, Supplementary Fig. 14f).

**Horizontally acquired pathogen-induced genes were largely inherited from a common ancestor**

Functional similarities between the responses of *A. ricciae* and *A. vaga* might either reflect shared inheritance of an ancient HGT_C portfolio common to many bdelloid rotifers, or independent gains of genes with common functions but different and potentially recent non-metazoan origins. We tested these alternative hypotheses by examining patterns of presence or absence for all upregulated HGT_C across *A. vaga*, *A. ricciae*, and other sequenced bdelloid genomes (see Supplementary

Methods). Of 372 $HGT_C$ upregulated by *A. ricciae* at T24, 351 (94%) showed evidence (Diamond 'blastp' hit with *E*-value < 1e−5) of orthologs in distant bdelloid genera (*Rotaria* or *Didymodactylos*), consistent with acquisition millions of years ago[93] (Supplementary Data 12). All the remaining 21 genes shared orthologs within *Adineta*, indicating gains at least as old as congeneric divergences. Similarity between the two focal species, therefore, seems to reflect inheritance of horizontally acquired genes from a common ancestor. In *A. vaga*, at T24, just one upregulated $HGT_C$ out of 285 (AVAG|g23567, marked with an asterisk in Fig. 5a) had no identified orthologous counterparts and represents a putative recent acquisition. Consistent with this hypothesis, its corresponding gene model lacks introns and shows a marked elevation in GC content relative to its genomic background (Supplementary Fig. 15). It appears to encode an NRPS-PKS hybrid synthetase with closest UniProt/Swiss-Prot matches to bacterial PKS that synthesise macrolide antibiotics such as methymycin and pikromycin (Supplementary Data 8). The SeMPI prediction for this cluster is a small linear polyketide whose closest named match is alpiniamide (metabolite score 0.66), a product of *Streptomyces* sp. whose activity is unknown (Supplementary Data 11).

## Discussion

Although the scale and diversity of horizontally acquired genes encoded by bdelloid rotifers were recognised over 15 years ago[43], relatively little is known about their function, expression, or links to phenotypes. We found that these genes are markedly enriched in the response of bdelloid rotifers to a fungal pathogen, as predicted by the evolutionary hypothesis that relentless interspecific conflict is a key selection pressure favouring genetic transfer. Among the most strongly upregulated products are putative NRP/PKS genes apparently acquired and modified from bacteria or fungi, which use related genes to produce secondary metabolites with antimicrobial activity. Both bdelloid species encode dozens of biosynthetic clusters, but the more resistant species *A. ricciae* upregulates more of these, more rapidly and to a greater degree when attacked by *R. globospora* than does the less resistant *A. vaga*. Together, these results provide evidence for the hypothesis that bdelloid rotifers defend themselves against natural enemies in part by expressing horizontally acquired biosynthetic pathways used for interspecific conflict in other kingdoms. It is noteworthy that such an unusual strategy should have emerged in a class of metazoans lacking reports of males or mating[30].

Among eukaryotes, HGT is widely detected in fungi[94], including acquisition of NRP/PKS genes from bacteria, but is thought to be far rarer among animals[45,95]. While some putative horizontally acquired genes might conceivably represent novel gene families with a deep or obscure metazoan heritage, the horizontal origin of the genes we highlight is well supported by phylogenetic evidence. Indeed, two nonribosomal peptide synthetases of apparent bacterial origin were among the first non-metazoan genes to be detected in *A. vaga* genomic libraries, leading to an early suggestion that bdelloid biosynthetic activity might include the production of secondary metabolites[43]. However, NRP/PKS genes were not explored further until now, perhaps partly because of difficulties in assembling and annotating these large, multimodular and highly duplicated genes, and partly because of low levels of baseline expression.

That bdelloid rotifers encode and express so many apparently functional NRP/PKS is highly unusual. PKS genes are known from a few animals, some potentially gained via horizontal transfer, but others encoded by ancestrally metazoan genes that usually comprise a single module[96]. NRPS and multimodular PKS-NRPS are thought to be very rare in animals, with only isolated cases identified[84,85,97]. Some of these lack the canonical domains of fungal and bacterial NRPS and are associated with neuroregulatory functions rather than pathogen defence or secondary metabolism[98]. To our knowledge, the only comparable case described previously is the springtail *Folsomia*

*candida*, whose genome also has been estimated to encode an elevated proportion of $HGT_C$, including several putative NRPS genes whose functions are unclear[48,85,99].

A key question is what compounds are synthesised by these genes. The rotifer proteins show low similarity to even the best-matching bacterial hits, and initial secondary metabolite predictions indicate products with limited similarity to known compounds. However, the top hits for both proteins and products appear to match antimicrobial or cytotoxic compounds with activity against fungi, rather than pigments, siderophores or other diverse products of fungal and bacterial synthetases. The co-location and co-expression with metazoan and prokaryotic pumps that are known to export antibiotics and toxins out of cells[100] is also consistent with such a function. A key limitation is the current lack of genetic tools to test the function of genes of interest in bdelloids. Until such methods become available, functional and comparative analysis of differential expression data provides the best initial insight into genes that might mediate the pathogen response and its variation among species.

A role for the strong upregulation of $HGT_C$ RNA ligases in both species may be suggested by molecular mechanisms in other host-parasite systems. Close relatives of *Rotiferophthora* in the fungal order *Hypocreales* attack their insect hosts using ribotoxins as virulence factors[101–104]. These secreted ribonucleases cleave the highly conserved sarcin-ricin loop in the large ribosomal subunit of host cells, fatally inhibiting protein synthesis[102]. If *Rotiferophthora* hyphae secrete similar virulence factors against *Adineta*, then rapid upregulation of proteins with RNA ligase activity might help the host to react by protecting or repairing ribosomes[105]. According to a recent review, RNA-targeting toxins, RNA ligases and related RNA repair systems are "extensively disseminated by lateral transfer between distant prokaryotic and microbial eukaryotic lineages consistent with intense inter-organismal conflict"[105]. This raises the hypothesis that bdelloids have borrowed molecular components from RNA-based microbial warfare to defend against an RNA-targeted attack by their own pathogens.

Both species also upregulated carbohydrate-active enzymes of bacterial and fungal origin with a range of glycoside hydrolase family domains. The function of these and other upregulated glucan-binding enzymes in pathogen defence is unclear, but two GH16 family genes acquired horizontally from fungi by a nematode have been linked to digestion of fungal cell walls and inhibition of conidial germination[92]. Upregulation of similar genes by bdelloids could mediate resistance by targeting components of fungal cell walls such as chitin or glucans, either as recognition receptors or by degrading them directly[106–108]. Glucanases are secreted by plants and bacteria as antifungal compounds[109,110], mediate antagonism between fungi[111], and are implicated in insect antifungal defences against *Metarhizium*, a close relative of *Rotiferophthora*[107].

The stringent thresholds we set here focus attention on the most strongly upregulated sets of genes. Relaxation of these thresholds would reveal additional genes of interest but is unlikely to affect the prominence of three major functional classes: NRP/PKS biosynthesis, RNA repair, and glucan binding. We focused on $HGT_C$ genes here and did not closely explore differential expression of metazoan genes within or between species, so it remains to be seen whether $HGT_C$ effectors replace or complement functions typically provided by the innate immune system[53].

The hypothesis that $HGT_C$ expression is important in pathogen defence might initially seem to predict stronger enrichment among upregulated versus downregulated genes. Although upregulated $HGT_C$ genes outnumber downregulated in every condition, enrichment was proportionally similar in both directions. One explanation— that $HGT_C$ are more loosely regulated and fluctuate more in expression than metazoan genes—can be rejected because desiccating rotifers showed no such pattern. A second possibility is that rotifers often regulate $HGT_C$ products using $HGT_C$ regulatory proteins, either

because these were acquired together or became 'wired' together in their new metazoan context. If so, then upregulation of $HGT_C$-enriched effectors might be paired with downregulation of $HGT_C$-enriched negative regulators or repressors. This explanation has some support: the word "regulation" appears in 40% of GO terms enriched among $HGT_C$ that are downregulated in response to pathogens, but not at all among upregulated genes (Supplementary Table 8). Thirdly, perhaps $HGT_C$ genes performing other functions are downregulated to divert resources to the immediate threat posed by the fungus. This could include genes with functions unrelated to immune defence, or constitutively expressed against bacteria present in the cultures or general microbial attack. For example, in nematodes, induction of antifungal defences correlates with repression of antibacterial immunity genes[112], perhaps to focus resources on a pressing threat or balance biochemical trade-offs in defence mechanisms[113]. This hypothesis could help explain why, at T24, *A. ricciae* significantly downregulated five $HGT_C$ NRP/PKS CDS (Fig. 5d) that had been substantially expressed under control conditions. The magnitude of this downregulation was higher than the single downregulated NRP/PKS CDS in *A. vaga* (Fig. 5d, Supplementary Data 7), consistent with the hypothesis that *A. ricciae* requires more downregulation of constitutively expressed NRP/PKS to match stronger upregulation of the focal cluster. $HGT_C$ enrichment among downregulated genes might therefore reflect trade-offs among pathogen-specific defensive $HGT_C$ sets, combined with pairing between $HGT_C$ regulatory and effector genes.

An association between horizontally acquired genes and bdelloid defence against pathogens is consistent with a range of evidence that genetic transfer may be evolutionarily favoured in part to address biotic conflict, and with the specific hypothesis that diseases pose a challenge for lineages where sex is rare or absent[2,114], with special measures required to keep up in the long term[21,115,116]. Of hundreds of differentially expressed $HGT_C$ we identified, the vast majority were gained prior to the common ancestor of *A. vaga* and *A. ricciae*, with the interesting potential exception of one NRPS-PKS hybrid cluster in *A. vaga*. Most are shared even more deeply among bdelloid rotifers. Gene gains and losses, therefore, do not seem to occur in contemporary bdelloid species at the speed or scale theoretically required to sustain rapid coevolution, or to be comparable with sexual recombination or prokaryotic HGT dynamics[117–122]. Over deep evolutionary time, however, bdelloids have built up a biochemical repertoire that could provide enhanced defensive capabilities and the raw material for genetic differences to arise.

One feature that could promote variability over shorter timescales is the enrichment of NRP/PKS genes in subtelomeric regions. These tend to be dynamic, and in bdelloids harbour a high concentration of transposable elements, including the giant *Terminon* elements[123], which can facilitate translocation. A previous study in *A. vaga* provided evidence that an NRPS gene is mobilised within the genome in association with transposable elements[124]. A dynamic genome location increases the opportunities for duplication, intragenomic mobilisation and recombination of the modular genes encoding these complex enzymes[125,126]. While the dynamic nature of subtelomeric regions makes it hard to assemble these regions robustly, the comparison between *A. ricciae* and *A. vaga* provides initial support for this model: an $HGT_C$ cluster initially shared by both species has apparently undergone serial duplications or losses between them, probably contributing to the markedly different expression patterns we detected. If confirmed by further investigation, including in other bdelloid species, these mechanisms would represent an unusual avenue of defensive evolution for an animal.

Rare gain of new antimicrobial pathways, coupled with ongoing rearrangement and expansion of acquired clusters in dynamic genome regions, might help in the long term to diversify defensive phenotypes among bdelloid lineages. Further comparisons of resistance and gene expression among different host and pathogen species will be needed

to test for such differences, and to interpret their links to coevolutionary theory[14,127]. In the shorter term, ecology plays a key role: bdelloid rotifers can escape from epidemics of *R. globospora* and other pathogens by dispersing in a desiccated state that the fungus does not tolerate[59,61]. This dispersal asymmetry could help host populations persist long enough to evolve physiological resistance via the longer-term processes posited here, while reducing reciprocal opportunities for a pathogen to evolve resistance to a repeatedly encountered antimicrobial compound. Together, these unusual ecological and genetic factors might help explain how these rotifers keep pace with coevolving antagonists despite the long-term rarity or absence of sexual outcrossing.

Finally, our results raise the intriguing possibility of discovering new antimicrobial compounds in bdelloids. Prospecting for antimicrobials in nature is largely limited to bacteria and fungi. Among various barriers to developing successful products is the high probability that a compound will be toxic to animal cells and, therefore, fail early stages of testing[128]. Our results suggest that bdelloid rotifers have spent millions of years 'bioprospecting' for antimicrobial synthesis machinery across multiple domains of life, and adapting it for large-scale expression in animal cells to defend against fungi or bacteria. If so, investigations of the secondary metabolome of bdelloid rotifers[129] may be of considerable interest in the search for novel antimicrobial agents to treat animal infections.

## Methods

### Rotifer and pathogen isolates
Animals belonging to the species *Adineta ricciae*[69] and *A. vaga*[130,131] were isolated respectively in 1998 from mud in Australia[69] and ca. 1984 from moss in Italy[132]. The fungal pathogen *Rotiferophthora globospora*[68] was found attacking co-occurring rotifers of the genus *Adineta* in soil in northern New York[60]. A pure culture on potato dextrose agar (PDA) was obtained in December 2008 using methods described elsewhere[133], and deposited in April 2009 with the USDA Agricultural Research Service Collection of Entomopathogenic Fungal Cultures (ARSEF) for long-term cryogenic storage under the accession code ARSEF 8995.

### Infection assays
Adult rotifers were transferred to 96-well plates (Thermo-Fisher), with approximately 11 animals per well (mean: 11.0, SD: 3.5) in 60 μL of sterilised, distilled water. Wells were inoculated with 8 μL of freshly prepared *R. globospora* conidial suspension at a density of 125 spores μL$^{-1}$. Negative control wells received 8 μL of distilled water or inactivated spore suspension (see Supplementary Methods). The final density of spores in each well was high (ca. 15 conidia μL$^{-1}$) to ensure every animal was exposed to the pathogen in a synchronised pulse. Rotifers were counted after 72 hours (Supplementary Fig. 16) and classed as active, contracted, killed by infection (if at least one hypha had emerged through the integument from the interior) or otherwise dead.

### RNA-seq experimental design
Rotifer populations were reared in eight replicate Petri dishes per species, with ca. 50 founders per dish, fed only with *E. coli* (OP50, 5 ×10$^8$ cells per dish) in distilled water. Rotifers were counted and harvested after 4 weeks, when the mean population size was about 3000 per dish for *A. ricciae* and 2000 for *A. vaga*. Populations were then subdivided to yield 16 replicate 1.5 mL tubes for each species, with approximately 1000 animals per tube for *A. ricciae* and 600 for *A. vaga*. Tubes were then randomly allocated to receive either live or irradiated pathogen spores, and to have RNA extracted either 7 or 24 hours later, with each combination replicated four times. Irradiated spore suspensions (see Supplementary Methods) were used as a control treatment to account for physical, chemical, or nutritional effects of ingesting fungal cells, so that all else was equal except for pathogen viability. Tubes were

inoculated with 20 μL of live or irradiated spore suspension at a density of 500 spores μL$^{-1}$, for a total of 10,000 spores and a final density of 80 conidia μL$^{-1}$, to ensure every animal was exposed as synchronously as possible, and then incubated upright at 20 °C in a blocked layout until RNA extraction.

## RNA extraction and sequencing
Total RNA was extracted from each tube at the appropriate time-point using a column-based RNeasy Mini kit (Qiagen #74104), following the manufacturer's protocol for animal tissues. Extracted RNA was eluted in 32 μL of RNase-free water and 1.5 μL aliquots were analysed using a Nanodrop 2000 (ThermoFisher). Spectro-photometric measurements were used to select the three replicates with the highest RNA concentrations from each treatment group for further analysis and sequencing (24 tubes total). Libraries were prepared using the TruSeq stranded mRNA kit (Illumina) and sequenced on an Illumina NovaSeq 6000 at Edinburgh Genomics (Edinburgh, UK), using an SP flow cell to generate 50-base paired-end reads (-200 bp inserts).

## Data filtering and quality control
Raw sequencing reads were quality- and adapter-trimmed using BBTools 'bbduk' v38.73 and error-corrected using BBTools 'tadpole' (https://sourceforge.net/projects/bbmap/). Unwanted ribosomal RNA (rRNA) reads were removed by mapping to the SILVA rRNA database[134] using BBTools 'bbmap'. Contaminant reads derived from either the bacterial rotifer food present in the dishes (*E. coli* strain OP50) or from the fungal pathogen itself were removed using a similar approach, by mapping to the OP50 genome (NCBI accession GCF_009496595.1) or to all available genomes of fungi in the family *Clavicipitaceae* (NCBI taxid 34397; see Supplementary Methods for further details). An average of 78.5 million reads were retained per library after filtering (94.2 Gb total data; Supplementary Table 9), with >99% of filtered data mapping to the *A. vaga* (Av13) and *A. ricciae* (Ar18) reference genomes (Supplementary Table 10). We tested for and excluded any residual contribution of RNA reads from contaminating bacteria to differential expression calculations for significantly upregulated HGT$_C$ (Supplementary Tables 11–15).

Code and source data for core bioinformatics analyses are available online at https://doi.org/10.5281/zenodo.11402163[135].

## Differential expression and functional enrichment analyses
Transcript quantification was performed using Salmon 'quant' v0.14.1[136], using the gene models of Nowell et al. (2018)[37] as the target transcriptomes. Short transcripts (< 150 bases) were removed prior to analysis, and genomic scaffolds were appended to each transcriptome as 'decoys' prior to quantification as recommended in the Salmon documentation. Relationships between biological replicates within and between samples were checked visually using utility scripts from the Trinity software[137], with results from PCA indicating high correlation in gene expression among replicates (Supplementary Fig. 17). Statistical analysis of the resulting count matrix was performed with DESeq2 v1.26.0[138], which uses negative binomial generalised linear models to test for differential expression. *P*-values were adjusted for multiple testing using the Benjamini-Hochberg method[139] to control the false discovery rate (FDR). Stringent thresholds of FDR <1e−3 and absolute log$_2$ fold-change > 2 (i.e., 4-fold difference in expression) were used to define an initial set of differentially expressed genes for downstream analysis. Comparisons of control populations showed that HGT$_C$ genes are expressed at significantly lower levels than non-HGT$_C$ genes, indicating that the enrichment for HGT$_C$ among differentially expressed genes is unlikely to be explained by a known bias[140,141] toward genes with higher expression levels (Supplementary Figs. 18 and 19). To test whether our results were affected by the

choice of significance thresholds used or DE software, we repeated the analysis using a range of fold-change thresholds (1.5-, 2-, 8-, and 16-fold absolute differences yielded the same results as 4-fold) and across two alternative DE packages, with consistent results (Supplementary Figs. 5–10; see Supplementary Methods for further details). At the 1.5-fold threshold, the proportion of all HGT$_C$ that were differentially expressed at T24 was 19.7% for *A. ricciae* and 11.5% for *A. vaga*.

Functional annotations for all genes were assimilated using the Trinotate v3.2.0[142] pipeline, while putative HGT candidate genes (HGT$_C$) were classified based on the analysis of Jaron et al. (2021), and cross-checked with the recent publication of a chromosome-level assembly for *A. vaga*[40] (see Supplementary Methods for further details). Functional enrichment analysis was performed using GOseq v1.38.0[75], based on gene ontology (GO) terms identified during functional annotation. For each timepoint, the test set was defined as HGT$_C$ genes that were significantly up- or downregulated (based on absolute fold-change > 4 and FDR < 1e−3) versus the background set of the whole genome. Significant GO terms were visualised using Revigo[143] (default parameters).

To test whether our results were affected by the choice of genome-based transcriptome targets used, we repeated the above analyses using transcriptomes that were assembled de novo from the RNA-seq data (see Supplementary Methods for further details), independent of reference assemblies or HGT$_C$ filtering (Supplementary Data 4). We found 'phosphopantetheine binding' (GO:0031177) was still the most highly enriched term for *A. ricciae* (FDR = 3.72e−15), with 'antibiotic biosynthetic process' and 'isomerase activity' also significantly enriched (FDR = 0.0001), whereas none of these NRP/PKS-associated terms was enriched at any level for *A. vaga*. The same conclusion is therefore reached whether transcriptomes are assembled and annotated de novo from RNA-seq data or mapped to gene models predicted from the reference genomes.

To account for the lower RNA-seq coverage and power to detect functional enrichment in the desiccation dataset[51], we relaxed the threshold for GO term enrichment to FDR < 0.05, then to FDR < 0.1 (Supplementary Data 5). This ensured we would not miss weak signals of functional overlap between HGT$_C$ upregulated in response to biotic and abiotic stress.

## Statistical analyses
Differences in infection mortality between the species were calculated as relative risk using the 'riskratio' function from the fmsb package (v0.7.6) in R v4.2.0, testing for independence between species identity and infection. To assess differences in transcriptional response to infection between species, we fitted a linear mixed effects model to predict DE (log$_2$ fold-change) including species and timepoint as two-level fixed factors, DE category ('upregulated', 'downregulated' and 'non-significant' according to the thresholds outlined above) as a three-level fixed factor, and gene ID as a random intercept term using the 'lmer' function from the lme4[144] package in R v4.0.2[145]. To assess the significance of differences in the proportion of HGT$_C$ in up- and downregulated subsets of genes, we first fitted generalised linear models for HGT$_C$ as a binary response variable (1 = is HGT$_C$; 0 = is not HGT$_C$) against species as a two-level fixed factor and DE category as a three-level fixed factor using the 'glm' function in base R. Separate models were fitted for T7 and T24 timepoints. A reduced model was then fitted using 'glmer' from lme4, including DE category as a three-level fixed factor and gene ID as a random intercept term. Frequency-based tests of statistical significance were calculated on the two-tailed basis.

## Putative NRP/PKS screen and genomic validation in *A. vaga*
An automated screen was conducted on the predicted proteomes of *A. vaga* and *A. ricciae* for putative NRP/PKS genes based on the

presence of the canonical adenylation (AMP-binding, Pfam accession PF00501), thiolation and peptide carrier protein (PP-binding, PF00550) and condensation (PF00668) domains, using HMMER 'hmmsearch' v3.3 (http://hmmer.org/). Only proteins with significant matches ($E$-value $\leq 1e{-}5$) to all three domains were classified as putative NRP/PKS in this 'three-domain' set of gene models. This approach was validated by performing a manual survey in *A. vaga*, where a chromosome-scale genome assembly (Av20) has recently become available[40] (Supplementary Fig. 12; see Supplementary Methods for further details).

To test whether the identified NRP/PKS set were located in telomeric regions, we counted the frequency of (*i*) genes, (*ii*) transposable elements (TEs)[38], and (*iii*) the telomere-associated repeat motif 'TGTGGG'[146] in (max) 50 kb windows surrounding each putative NRP/PKS gene model using BEDTools v2.29.2. The span of each feature was converted to a proportion by dividing by the actual window size for each flanking region, to correct for variation in window size. The genomic context of core eukaryote BUSCO genes ($n = 303$) was also evaluated for comparison (Supplementary Fig. 13 and Supplementary Table 6). Boxplots in Fig. 5d, e show median, interquartile range (IQR), and whiskers extending to the farthest datapoint from the median that is within 1.5*IQR of Q1 and Q3 respectively; points beyond whiskers are statistical outliers. For *A. vaga*, the genomic locations of NRP/PKS clusters were also ascertained directly based on the Av20 genome assembly. The Av20 assembly (along with an additional, alternative assembly for *A. ricciae*) was also used to determine copy number and orthology of the set of upregulated putative NRP/PKS gene models highlighted in Fig. 5 (see Supplementary Methods for further details).

### Phylogenetic analyses

Phylogenetic trees of rotifer-encoded NRP/PKS proteins with bacterial and fungal homologues were constructed based on the alignment of the condensation domain (PF00668) with Pfam 'seed' representatives, best-matching homologues from Uniref90 searches, and selected known occurrences of NRP/PKS in animals[85] (see Supplementary Methods for further details). Alignments were built using HMMER 'hmmalign', and phylogenetic analysis performed using IQ-TREE v1.6.12[147]. A similar approach was used to construct phylogenies of other domains of interest linked to our GO enrichment analysis of upregulated HGT$_C$ (Supplementary Fig. 14), including various RNA ligase domains (PF13563, PF09414, PF09511); the glycosyl hydrolase domains Glyco_hydro_16 (PF00722) and Glyco_hydro_64 (PF16483), and the caspase domain Peptidase_C14 (PF00656). This last domain was plotted to contextualise an upregulated gene (ARIC|g44127) matching a metacaspase of apparent fungal origin; the HMM search revealed 101 and 69 proteins with a significant match to the Peptidase_C14 domain in *A. vaga* and *A. ricciae* respectively. Many of these were marked as HGT$_C$ from bacteria or fungi but are highly divergent from known caspase homologues in either group, indicating a bdelloid-specific expansion. Only a small number were found to be significantly differentially expressed on exposure to the pathogen, mostly downregulated ($n = 18$ and 8 for *A. vaga* and *A. ricciae* respectively; 4 and 2 upregulated).

### Secondary metabolite prediction

Secondary metabolite products of focal NRP/PKS gene models were predicted using the SeMPI v2.0 web server[87], with maximum cluster distance set to 25 kb and all metabolite databases selected, but otherwise default settings (see Supplementary Methods for further details).

### Reporting summary

Further information on research design is available in the Nature Portfolio Reporting Summary linked to this article.

## Data availability

All raw sequencing data generated by this study have been deposited in the relevant International Nucleotide Sequence Database Collaboration (INSDC) database with the BioProject ID PRJEB39927 and the SRA run accessions ERR4469891, ERR4469902–8, ERR4471099–102, ERR4471104–11, and ERR4471113–6 (see Supplementary Table 9). This study also analysed publicly available data from BioProjects PRJEB1171, PRJEB23547, PRJEB43248, and PRJNA494578. Source data are provided with this paper.

## Code availability

All custom scripts, source data, and bioinformatics workflows used in this study are published in a publicly available GitHub digital repository (https://doi.org/10.5281/zenodo.11402163).

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

## Acknowledgements

Transcriptome sequencing was performed by the UK Natural Environment Research Council (NERC) Biomolecular Analysis Facility at Edinburgh Genomics at the University of Edinburgh (NBAF-Edinburgh), and we acknowledge the valuable services of the USDA ARS Collection of Entomopathogenic Fungal Cultures (ARSEF). The authors wish to thank Matthew Meselson, Thomas Lanyon-Hogg, Alan Tunnacliffe and three reviewers for valuable comments; Matthew Arno, Colin Sharp and Rebecca Allen for sequencing support; Juli Cohen, Isobel Eyres, Chiara Boschetti, and Mariya P. Dobreva for RNA extraction advice; and the Ashworth Compute Co-operative Cluster (AC3) at the Institute of Ecol-ogy and Evolution, University of Edinburgh. This work was funded by NERC Fellowship NE/J01933X/1 (C.G.W.); EMBO Long-Term Fellowship 733-2010 (C.G.W.); a 2012 award by The Gen Foundation (C.G.W.); NERC

grant NE/M01651X/1 (T.G.B.); NERC grant NE/S010866/1 (T.G.B., R.W.N. and C.G.W.); NIH grant R01GM111917 (F.R. and I.A.) and NIH NIA R21AG046899 (D.B.M.W.).

## Author contributions

Conceptualisation: C.G.W., T.G.B.; methodology: C.G.W., R.W.N., T.G.B., F.R., I.A.; experimental investigation: C.G.W.; software: R.W.N., F.R.; validation: R.W.N., C.G.W., F.R., B.H.L., D.B.M.W., I.A.; formal analysis: R.W.N., C.G.W., T.G.B.; resources: C.G.W., R.W.N., F.R., B.H.L., D.B.M.W.; original drafts: C.G.W., R.W.N., T.G.B., review and editing: T.G.B., C.G.W., R.W.N., I.A., D.B.M.W., F.R.; visualisation: R.W.N., C.G.W., F.R.; supervision: T.G.B., C.G.W., I.A., D.B.M.W.; project administration: C.G.W., T.G.B.; funding acquisition: C.G.W., T.G.B., R.W.N., I.A., D.B.M.W.

## Competing interests

C.G.W. and T.G.B. are inventors on an Oxford University Innovation priority patent application (GB N427450) relating to this paper; otherwise all authors have no competing interests.
