## [Peer Review File · Nature Communications]

Bdelloid rotifers deploy horizontally acquired biosynthetic genes against a fungal pathogenREVIEWER COMMENTS

Reviewer #1 (Remarks to the Author):

In this manuscript Nowell et al. examine the transcriptional responses of two bdelloid rotifer species to fungal infection, focusing on the large number of horizontally transferred genes found in this fascinating group of animals. The two species of bdelloids under investigation here show differential susceptibility to infection by the fungus *Rotiferophthora globospora*: *Adineta vaga* shows high mortality post infection, whereas *A. ricciae* does not. The authors performed RNA sequencing on both species at two timepoints post-infection (7 and 24 hours) and compared gene expression changes to animals exposed to uv-killed spores. They found that the horizontally transferred genes showed enriched differential expression post-infection; this enrichment was not observed in response to desiccation. By comparing gene ontology category enrichment between the fungus-susceptible and fungus-resistant species, the authors uncovered a striking upregulation of genes required for non-ribosomal peptide/polyketide synthetase (NRP/PKS) activity in *A. ricciae* (the fungus-resistant species). The authors characterized the NRP/PKS genes in both species and found a clade whose expression was upregulated in response to infection, which also showed an increased copy number in *A. ricciae* relative to *A. vaga*, and that were flanked by multiple copies of transporter-encoding genes associated with antibiotic resistance and toxic compound export in other systems.

This work implicates horizontally transferred genes in bdelloids with their responses to pathogens and provides several testable hypotheses that should drive future work. Thus, I believe that this paper provides a significant advance that merits publication here. The work is carefully and rigorously conducted, and thoughtfully presented. I have only a few relatively minor suggestions/comments that the authors should consider incorporating in a revised version.

1. Fig. 1c shows the differences between *A. vaga* and *A. ricciae* gene expression based solely on the number of differentially expressed genes. It may also be helpful for the authors to include principal component analysis to provide another overview of the differences and similarities between the post-infection time points and their respective controls for each species.

2. When incorporating the RNAseq analysis from desiccation/rehydration, the authors correctly note the need for caution in comparing datasets derived from different experiments. It seems likely that the authors' conclusions about the differences between the response to infection and to abiotic stress are correct. However, since the authors don't explain the time points used for the desiccation/rehydration experiments, it remains unclear if these differences could instead be due to looking at completely unrelated time points relative to the onset of stress.

3. The criteria for choosing the post-infection time points are spelled out clearly in the Supplementary Information (lines 186-190), but I didn't see them in the main text. This information helps readers understand the rationale for these experiments, so it seems that the audience would be best served by including these criteria in the main text.

4. Similarly, many readers will immediately note that the horizontally transferred genes are proportionately up- and down-regulated (i.e., there is no stronger enrichment of upregulated vs. downregulated genes). The authors provide thoughtful discussion of why this might be the case in a Supplementary Note. Not every reader will take the time to go through a 45-page supplement, so it seems a shame not to include these thoughts in the main Discussion instead.

Reviewer #2 (Remarks to the Author):

Dear Editor, Dear authors,

Title: Bdelloid rotifers deploy biosynthetic genes acquired from bacteria against attack by

pathogenic fungi

This study meticulously delves into the presence/roles of horizontally transferred genes (HGT) in the coevolutionary defense mechanisms of bdelloid rotifers against their pathogenic adversaries, particularly the fungus *R. globospora*. Using a combination of infection assays, novel and pre-existing transcriptomic datasets, and comprehensive bioinformatics techniques (functional prediction tools; phylogeny; ...), the authors offer valuable insights into the puzzling evolutionary persistence and significant fraction (+-10%) of HGT within bdelloid genomes.

Past studies have highlighted the prevalence of HGT in bdelloid species. However, the processes by which these genes were acquired and the evolutionary rationale for their retention in the genome are not entirely clear. Several hypotheses have been advanced, but empirical, particularly functional, data to support these theories is limited.

In this paper, two related species (and morphologically "very similar"), *Adineta vaga* and *A. ricciae*, were chosen to investigate transcriptional changes following fungal exposure. It was observed that *A. ricciae* demonstrated a higher resilience to the fungal infection when compared to *A. vaga*. To explore this further, transcriptional responses at 7 hours and 24 hours post-infection were examined. These time points consistent with microscopic and infection assay data, indicated changes in the transcriptomes of both species. A significant finding was the enrichment of HGT in differentially expressed genes, which might be involved in the rotifers' response to infections. Additionally, when this data was compared with prior transcriptomic studies on *A. vaga*'s desiccation and rehydration, the study highlighted the distinct roles HGTs might play in different physiological scenarios. Then, a Gene Ontology analysis of differentially expressed HGTs suggested specific patterns in *A. ricciae* that could be linked to its fungal resistance. Further examination identified a higher occurrence of NRP/PKS proteins in *A. ricciae* than in *A. vaga*. Predictive tools used in the study suggest that these genes might have a role in producing compounds similar to known antibiotics. RNA ligases and glucanases were also identified in *A. ricciae* as candidate genes for pathogen resistance. The paper concludes with the hypothesis that the HGT associated with pathogen defense might have been inherited from a common bdelloid ancestor, a suggestion supported by orthologous studies across various bdelloid species.

The manuscript is structured with commendable coherence, demonstrating a logical progression that effectively anticipates and addresses the reader's queries. Any questions arising at the end of a section are adeptly answered in the subsequent one. The meticulous details in the Methods & Materials, along with the supplementary data, underscore the authors' rigorous approach throughout the experimentation process. They have evidently taken measures to account for potential biases. This thoroughness not only underscores the authors' professionalism but also instills confidence in the presented results.

While earlier research primarily highlighted the substantial presence of HGT in bdelloids or focused on a select few genes possibly acquired via HGT, this study stands out as the first to offer compelling evidence that HGT plays a pivotal role in the evolution and sustenance of asexual bdelloids, especially in their response to pathogens. Although the study lacks genetic tools for functional validation – a limitation candidly acknowledged by the authors – the robustness of the data presented substantiates the core thesis put forth by the authors, within the confines of the current knowledge and tools available for bdelloid rotifers.

Below are some comments that could enhance the manuscript. From my perspective, these improvements can make the content more accessible to readers unfamiliar with bdelloids, streamlining comprehension without compromising the quality of the work.

I would suggest the authors to not hesitate to provide some extra definitions in their manuscript to clarify the context of the paper and ensure easy access to scientific audience. For example, line 75 authors refer to parthenogenetic eggs. A short definition of this association may be relevant.

On Line 81: Consider including the recent publication by Terwagne et al. (2022) that explores the occurrence of a modified meiotic process in bdelloid germ cells. This work suggests that reproduction isn't strictly clonal in these organisms: Terwagne, M., Nicolas, E., Hespeels, B., Herter,

L., Virgo, J., Demazy, C., Heuskin, A.C., Hallet, B., & Van Doninck, K. (2022). DNA repair during nonreductional meiosis in the asexual rotifer *Adineta vaga*. *Science Advances*, 8(48), eadc8829. doi: 10.1126/sciadv.adc8829.

On Line 97: For a more neutral presentation of your hypothesis, consider omitting the word "particularly." While the introduction does provide evidence suggesting a role of HGT in the bdelloid immune system, prior to this study, there was no basis to anticipate whether the bdelloid genome might have an enriched fraction of HGT or merely a handful of such genes.

On Line 106: The use of the term "disproportionately" seems repetitive and might come across as overly emphatic in various sections of the manuscript. While it's clear from the paper that there's a notable enrichment of HGT in the described immune response, a more neutral phrasing might be more appropriate. Given that we are concluding the introduction section, consider referencing a "potential enrichment" or "increased likelihood" instead or better alternative to find by the authors. On line 122: see my reference line 106.

On line 134: I was unable to find how author were able to conclude that >95% of animals ingested spores within minutes of exposure. Please clarify in the M&M section how it was reported.

On Line 138: The origins of *A. vaga* from moss in Italy and *A. ricciae* from a billabong in Australia, as detailed in the supplementary methods, suggest distinct ecological environments for the two species. Labeling these as similar might be misleading. The moss habitat of *A. vaga* can be characterized as limno-terrestrial, whereas *A. ricciae* was found in a pond environment, implying notable ecological differences. I'd recommend omitting this similarity claim or providing additional reasoning to substantiate the comparison.

Furthermore, this line prompts me to question the choice of models used. Why were these two related species specifically selected? Can we deduce the timeline of their divergence from common ancestors? Incorporating another bdelloid species might offer broader insights into both transcriptional responses and pathogen resistance. Exploring these responses in a more expansive context could either corroborate or challenge the findings of the current study. It might be valuable to incorporate these considerations either in the discussion or supplementary methodology sections.

On Line 168: The authors center their transcriptomic analysis on bdelloid HGT. Yet, even with the raw data accessible, there is a notable absence of information concerning the non-HGT response. While I grasp the authors' intent to concentrate on HGT, offering a concise overview of these results in the supplementary information could be beneficial for two main reasons: 1) to comprehensively document the immune response of bdelloids to fungi, and 2) to draw comparisons with the response specific to HGT. This point is addressed at the end of the manuscript. If there's no anticipation of featuring these details in a subsequent publication, I recommend (but not mandatory) that the authors incorporate (even a brief overview) them here for a more comprehensive understanding.

On Line 171: Can you confirm here that control populations were in contact with inactivated fungi?

On Line 186: Check Suppl Fig 2. Please italicized species name. Do for whole document.

One Line 185: I would remove the part "as expected if defensive or recognition pathways are conserved" as we have here no evidence at this stage that the differentially expressed genes are here involved in defensive or recognition pathways. Please adapt end of the section accordingly.

On Line 193: As outlined in the Methods & Materials section, the authors diligently validated their HGT using a recent and stringent methodology. However, when juxtaposed with previously published data, I noticed that the *A. vaga* genome in Simion et al., 2021 indicated an approximate 8.3% HGT. While the disparity between the two studies isn't substantial, I'm curious: did the authors briefly investigate the potential impact of alternative annotations on their results?

On Line 200: A central argument in the paper, suggesting that HGT plays a role in the immune response of hydrated bdelloids when exposed to fungi, is the significant portion of post-infection overexpressed genes annotated as HGT. Nonetheless, it's noteworthy that HGT enrichment was also observed in downregulated genes across both time points and species. Additionally, as highlighted later in the study, a considerable proportion of HGT is specific to the rehydration process in *A. vaga*. This context indicates that while many HGTs seem to be involved in the bdelloids' immune response, others (also many genes) may be linked with desiccation resistance or other yet-to-be-determined functions. It would be beneficial if the authors could also reference, discuss and develop this in their manuscript, encouraging further investigation into the multifaceted roles of HGT in bdelloid rotifers.

On Line 255: ".." check typo error.

On Line 278, Fig.3.a: In the legend, I would recommend including a full description of terms such as "NS core" for clarity. Throughout various sections of the manuscript, it occasionally becomes challenging to trace the origin or context of specific gene numbers. For instance, if "NS core" refers to non-differentially expressed genes, there seems to be a discrepancy; the figure lists 57,379 genes under "NS core", while another table in the manuscript references a geneset of 35,079. Furthermore, part c, despite the evident effort, is somewhat complex to decipher. The presentation of data could benefit from additional clarity. Mixing raw numbers with percentages can be particularly confusing for readers, so a more streamlined presentation would enhance comprehension.

On Line 378: While certain details are furnished in the Methods & Materials section, the current presentation leaves ambiguities about how the data can be employed to discern the mentioned "40 clusters." My interpretation of Figure 5c does not align with this representation, and further clarity is needed for a holistic understanding of Supplementary Data 8.

I'd also like to use this comment to suggest that the authors furnish more comprehensive context with their supplementary data. In its current state, readers often encounter excel sheets that can be challenging to decipher due to the absence of adequate descriptions. Offering a clearer breakdown or accompanying explanations would significantly aid in interpretation and understanding.

May be I'm wrong but I think that complete differentially analysis of gene expression (excel sheet or Deseq2 output) for fungi response and desiccation is lacking (data are only available for HGT or restricted amount of genes).

On Line 443: In the manuscript, the authors emphasize the heightened resistance of *A. ricciae* compared to *A. vaga* and also demonstrate the associated transcriptomic response. However, in this section, where there's an attempt to functionally characterize candidate genes, the reference is predominantly to *A. vaga*. What prompted the decision to base the analysis on *A. vaga* rather than on the more resistant *A. ricciae*? It would be beneficial to readers if the rationale behind this choice is elucidated in the manuscript.

On Line 492: This section could benefit from further clarification to strengthen the assertions made. For instance, the reference to "other sequenced bdelloid genomes" remains vague (only short name/ not complete or codes available in suppl data). It would be helpful to the readers if more details about these genomes could be provided, preferably within the supplementary data descriptions. For example, in its current state, Supplementary Data 4 might suggest the ancient acquisition of HGT; however, without more explicit information in the manuscript or supplementary data descriptions, drawing the authors' conclusions becomes challenging. I recommend elaborating on this.

While checking the manuscript, I noted a recurring theme which could lead to some confusion: genes identified in *A. ricciae*, presumed to play a pivotal role in its resistance against the fungi, are also observed in *A. vaga* and other bdelloid species. This raises several questions: Why don't these genes confer similar resistance in *A. vaga* (or were not enriched)? How do these genes function in other bdelloid species? Why maintain these HGT in *A. vaga* if they doesn't provide adequate protection against the fungi? Are there any indications that *A. vaga* might be more resilient than *A. ricciae* when faced with different pathogens? How do other species fare in similar scenarios? Addressing some of these points in the discussion section could enrich the narrative and insights of the article.

On Line 521: I'd like to refer back to the earlier mention about the use of "disproportionate." Considering that there is evidence that other HGTs might also be enriched in the rehydration response, it's crucial to tread carefully. The manuscript, in its current state, might inadvertently lead readers to believe that the primary driver for HGT acquisition in bdelloids is solely interspecific conflicts. Yet, a significant portion of HGT was not linked to pathogen responses. It might be enlightening at this juncture to present a more comprehensive overview of the data. For example: "Within the *A. ricciae* genome, X% of genes are identified as HGT. By t7, Y% of these genes exhibit overexpression. Of the differentially expressed genes, Z% are HGTs, with W% of total HGTs likely playing a role in the response to fungi." This broader perspective can offer a more nuanced understanding of the varied roles of HGTs in bdelloids.

Lines 621-624: I would suggest the authors to develop a little this part that may be obvious for bdelloid experts but not for a wide audience.

Supplementary Figure 11: Improve quality of figure.

Reviewer #3 (Remarks to the Author):

Review of the paper Nowell et al. for Nature Communications

Please accept my apologies for the delay in returning my review.

The presence of HGT in bdelloid rotifers is nowadays well-established and this manuscript highlights some additional evidence of the importance of horizontally acquired genes in the evolutionary adaptation of bdelloid rotifers. Previous studies have shown the role of some non-metazoan genes in the radiation tolerance and epigenetic regulation of bdelloid rotifers. This article of Nowell et al. shows that some of their antimicrobial mechanisms over-represented during fungal attack are of bacterial type. This is important in the field of the evolution of sex and coevolutionary antagonism, and more broadly in the field of antimicrobial resistances.

The transcriptomic analysis following pathogen exposure is solid: authors used three different packages to confirm the over-representation of HGTc in the DE genes and verified different fold changes (Supp Figs 3-8). They also compared with another study in which *A. vaga* was exposed to the abiotic stress "desiccation", and where the enrichment of HGTc was lower. I however have a general comment on this comparison within *A. vaga* (see below).

The authors then identified the putative functions of some HGTc that are only upregulated in the bdelloid resistant species *A. ricciae*, with some unique bacterial polyketide and nonribosomal peptide synthetases that seem to produce antibiotics.

More general comments:

It would have been interesting to include a timepoint at 7 days following infection to have an idea of the reproductive capacity following survival. *A. ricciae* indeed had a higher survival rate than *A. vaga* at 72h following fungal exposure, and the authors mention in the Supp data "By 72h, the proportion of infected animals in 125 experimental wells appeared to have stabilised. In a subset of wells recounted at 96h, only a 126 small fraction of animals (~6.5%) had newly developed visible infections." I wonder whether the *A. ricciae* population, triggering a strong resistant response at the start, i) was able to become pathogen-free after a few days or ii) did the population become weaker after several days and did the infection re-established? If the authors have some data on later timepoints, it would be interesting to include it in the supp data.

I am not so convinced by the difference observed in the transcriptomic response within *A. vaga* when comparing the biotic and abiotic stress. If you compare *A. vaga* biotic and abiotic response (Fig 4), is there a strong difference in functions between upregulated HGTc? In *A. vaga*, the upregulation of NRP/PKS functions was not detected, and while Table 1 shows a higher upregulation of HGTc when exposed to a biotic stress, the experiments were very different using *A. vaga* clones cultured in different laboratories and under different conditions. Therefore, can the authors highlight what are the main differences when comparing *A. vaga* exposed to biotic and abiotic stress, both for bdelloid-specific genes and for HGTc, including the putative functions identified? Some results are given for a few functions in the result section on RNA ligases and glucanases, a more detailed comparison is however required. This could be presented through a Venn diagram in Fig 4 for example (comparing b, c, d).

The title of the article should change, because this manuscript mainly highlights that more resistant bdelloid species deploy biosynthetic genes acquired from bacteria... than sensitive ones. It is not clear from the result section how similar is the functional response of *A. vaga* and *A. ricciae* to the fungal attack, and how different it is from the desiccation response within *A. vaga*.

To identify whether the NRP/PKS genes found in *A. vaga* and *A. ricciae* are restricted to fungi and bacteria (Fig 5), and which ones, or whether they are also found in other metazoans / eucaryotes (suggesting additional HGTc, see discussion lines 545-554), I advise the authors to provide the results of their BLAST analysis (BLASTp?) and to show the entire list of closely related sequences. The authors discuss this aspect in the discussion, without providing the entire list of blast results. The authors should also verify whether these genes are absent from monogonont rotifer genomes, a clade closely related to bdelloid rotifers, to confirm the acquisition within bdelloid rotifers specifically.

In this study, only two bdelloid species were studied, both being cultivated since decennia in laboratory conditions in the absence of pathogens. It would have been interesting to add a species sampled from the field, often exposed to pathogens. I wonder whether the expression profile is somehow biased, even in *A. ricciae*, because these species do not encounter pathogens since a long time. Nevertheless, the significant upregulation of HGTc when exposed to biotic stress in *A. ricciae* remains undeniable. Similar comment concerning a second pathogen. It would have been interesting to test more than one pathogen and confirm this upregulation when exposed to different pathogen lineages, to show it is a more common response to biotic stresses.

On a more speculative note, how would other parthenogens like oribatid mites and tardigrades, also living in soil and semi-terrestrial environments, deal with pathogens while HGT is not prevalent in these asexual species? Would long periods of desiccation be sufficient, as shown in previous papers of Chris Wilson? Why do bdelloid rotifers need to co-opt additional pathways from bacteria? It would be interesting to discuss this comparison with other parthenogens thriving in similar habitats.

More specific comments:

Introduction

- The paragraph (lines 50-53) on the antibiotic production in bacteria only focuses on NRPS and PKS, while it would be interesting here for the broader audience to give more examples.
- This sentence (lines 54-55) is not clear to me (because I am not a specialist) "... their mobility and modular structure facilitates diversification to produce a vast array of secondary metabolites...". What type of mobility and modular structure, and which secondary metabolites are produced. Some clarification here on this interesting aspect should be given.
- The authors mention on line 61, "Although meiotic shuffling has different effects to HGT, ...". Which different effects? It is a different mechanism to generate genetic diversity, but it is not clear which effect the authors refer to.
- Some seminal papers have recently been published on the reproductive mode of bdelloid rotifer *Adineta vaga*, showing a non-reductional modified meiosis I (Terwagne et al. *SciAdv*) and suggesting recombination (Vakhrusheva et al. *NatureComm* and Simion et al. *SciAdv*). Recombination has important consequences, and I am surprised the authors do not mention those latest results here in the introduction (paragraph on lines 74-79).
- In the paragraph on HGT (lines 81-87) authors should highlight that even desiccation sensitive bdelloid species, of the genus *Rotaria*, also have a high number of HGTs in their genome.
- On lines 88-91, HGT is suggested to be too slow to equate sexual shuffling seen in typical eukaryotes, UNLESS horizontal exchanges do occur between bdelloid individuals. While this is still debated, horizontal exchanges during desiccation could still be an important mechanism of generating diversity in this animal clade.

Materials and methods

- The infection assays were performed in sterilised distilled water WITHOUT food (line 649), such condition will induce the contraction of bdelloid individuals, even without pathogens. The authors mention in the results section (line 134) a rapid contraction of infected individuals, what about the negative control? Were there less contracted individuals when no pathogens were present in sterilised distilled water?
- In Supp Methods the authors mention (lines 86-88) "an individual host can resist the initial attack and prevent an ingested spore from establishing a successful infection, or at least delay its progression. Even partial resistance at an early stage could dramatically slow the spread of an epidemic...". How is the resistance visible, the spore is ingested but never germinates? Or the spore is not ingested, or? It would be interesting the authors provide more information on the phenotypic resistance observed, especially in *A. ricciae*.
- Putative HGTc (line 718) could have been classified as in the paper Simion et al. *SciAdv* using the Alienomics pipeline developed (see github). Alienomics considers the presence of other metazoan genes on the scaffold, besides other important parameters of HGT, rather than one gene of unambiguous metazoan origin as in Jaron et al. 2021 ("we considered non-metazoan genes as HGT candidates only if they were on a scaffold that also encoded at least one gene of unambiguous metazoan origin, to control for potential contamination in the genome assemblies")

Results

- Fig 1b. Points indicate replicate laboratory populations of *A. vaga* (red) and *A. ricciae* (blue); I count 20 blue points and 16 red points. In the M&M section it is mentioned "Adult rotifers were transferred to 96-well plates (Thermo-Fisher), with approximately 11 animals per well (mean: 11.0, SD: 3.5) in 60µL of sterilised, distilled water"
- Can the authors be more precise here, even in the Supp methods it is not entirely clear. Were there 16 replicates for *A. vaga* and 20 for *A. ricciae*? How are the 216 *A. ricciae* and 189 *A. vaga* represented in Fig 1b? Within each well (represented by a point on Fig 1b?) there was no variability in mortality between individuals? Is one point representing a population that increased in number (meaning it survived) or decreasing in number (meaning mortality)? If there was variability (some individuals died, other reproduced within the same well), how is this represented in Fig. 1b within each point?
- Fig 1c. At T24h there was a significant difference between *A. vaga* and *A. ricciae*, only for the upregulated genes, this could be indicated with an * on the figure.
- Lines 180-188: what is the overlap in transcripts between T7 and T24? How many genes found at T7 are also detected at T24, a Venn diagram could represent this overlap. Supp Fig 1 shows the DE correlation.
- Line 193: Jaron et al reported 10% for *A. vaga* and 10.6% for *A. ricciae*, Simion et al. reported 8.3% for *A. vaga*. What is this 11.5%?
- Fig. 4: it should be indicated in the title that Fig 4a and b represent the timepoint 24h for both species. It would be clearer if the authors present the results of Fig 4a and 4b in a Venn diagram to show what is common to both species in the upregulated HGTC response and what is specific to each species. Using colors and frames makes it less easy to interpret. Also compare the biotic and abiotic stress conditions only within *A. vaga* (see general comment above).
- Fig 5. a, the phylogeny could include more sequences of the blast analysis since NRP/PKS genes appear to be also known from a few animals (see discussion) and protists?

RESPONSE TO REVIEWERS' COMMENTS

Reviewer #1:

In this manuscript Nowell et al. examine the transcriptional responses of two bdelloid rotifer species to fungal infection, focusing on the large number of horizontally transferred genes found in this fascinating group of animals. The two species of bdelloids under investigation here show differential susceptibility to infection by the fungus *Rotiferophthora globospora*: *Adineta vaga* shows high mortality post infection, whereas *A. ricciae* does not. The authors performed RNA sequencing on both species at two timepoints post-infection (7 and 24 hours) and compared gene expression changes to animals exposed to UV-killed spores. They found that the horizontally transferred genes showed enriched differential expression post-infection; this enrichment was not observed in response to desiccation. By comparing gene ontology category enrichment between the fungus-susceptible and fungus-resistant species, the authors uncovered a striking upregulation of genes required for non-ribosomal peptide/polyketide synthetase (NRP/PKS) activity in *A. ricciae* (the fungus-resistant species). The authors characterized the NRP/PKS genes in both species and found a clade whose expression was upregulated in response to infection, which also showed an increased copy number in *A. ricciae* relative to *A. vaga*, and that were flanked by multiple copies of transporter-encoding genes associated with antibiotic resistance and toxic compound export in other systems.

We thank the Reviewer for this succinct summary of the major findings in our manuscript, clearly outlining the complementary lines of evidence that support the conclusions.

This work implicates horizontally transferred genes in bdelloids with their responses to pathogens and provides several testable hypotheses that should drive future work. Thus, I believe that this paper provides a significant advance that merits publication here. The work is carefully and rigorously conducted, and thoughtfully presented. I have only a few relatively minor suggestions/comments that the authors should consider incorporating in a revised version.

We are very glad to read these positive assessments especially in regard to the rigor and care involved in the analyses. We are grateful for the constructive suggestions and comments below.

1. Fig. 1c shows the differences between *A. vaga* and *A. ricciae* gene expression based solely on the number of differentially expressed genes. It may also be helpful for the authors to include principal component analysis to provide another overview of the differences and similarities between the post-infection time points and their respective controls for each species.

Many thanks to the Reviewer for this suggestion. We have added a new PCA plot to Fig. 1 (part c) that provides a graphical overview of total gene expression dynamics across timepoints and species. This complements the existing focus on strongly differentially expressed genes. The new PCA reveals a clear and consistent clustering of treatment (i.e. pathogen-exposed) groups from both species, relative to control groups. The replicates seem especially tightly clustered at timepoint T7, suggesting a shared response to fungal attack that is most similar during early stages of infection, then diverges somewhat between the more resistant and less resistant rotifer species. Results are discussed in the main text (see lines 153-161; 185). Additional PCAs that include the desiccation experiment groups are now included as part of the Supplementary Information (Supplementary Fig. 11, see also response below to Comment 2).

2. When incorporating the RNAseq analysis from desiccation/rehydration, the authors correctly note the need for caution in comparing datasets derived from different experiments. It seems likely that the authors' conclusions about the differences between the response to infection and to abiotic

stress are correct. However, since the authors don't explain the time points used for the desiccation/rehydration experiments, it remains unclear if these differences could instead be due to looking at completely unrelated time points relative to the onset of stress.

The infection experiment applied a large synchronised pulse of live (or inactivated) pathogen spores at timepoint T0 and tracked the progression of the response at 7h and 24h. We controlled for the timecourse within the pathogen experiment by measuring the non-infected populations at both timepoints. The desiccation experiment does not have timepoints in the same sense, because this is a gradual stressor by its nature, not one with a synchronised time of onset or the possibility of synchronised control groups. Groups were collected depending on their desiccation state rather than relative to a fixed T0. 'Hydrated' animals were collected after 2 days in tissue culture plates. 'Entering' animals were collected from similar plates between 2 and 4 days later, depending on when sufficient water had evaporated to leave a thin water film, with the animals contracted but still moving. 'Recovering' animals were left for about 5 more days to desiccate completely. The animals were left desiccated for 7 days, then were rehydrated and had RNA harvested 1 hour later. We now include this information on lines 262-264.

Given these details, trying to compare timepoints might not tell us much about relative stress patterns in the two experiments. Instead, Reviewer #1's first comment inspired us to conduct a PCA of total gene expression patterns, combining all replicates of the desiccation and pathogen stress experiments. We show this PCA in Supplementary Figure 11. The 'Entering' and 'Recovering' groups from the desiccation experiment are differentiated from their 'Hydrated' control group at least as strongly as the 'T7' and 'T24' replicates in the pathogen experiment are differentiated from their respective control groups, and in apparently similar directions with respect to the PC1 axis. This suggests to us that the stressors in the two experiments had similarly substantial effects on overall gene expression, a point we also make on lines 296-299 of the main text with respect to differentially expressed genes. The variation among the various control groups within the pathogen experiment (n = 12) appears similar in magnitude to the difference between the pathogen and the desiccation control groups, suggesting that the baseline transcriptional state of the animals was not dramatically different between the two experiments, relative to the changes induced by the respective stressors. We mention this analysis briefly on lines 300-302, as a potential source of evidence about the comparability between the two experiments.

While our revised manuscript continues to note the need for caution in comparing datasets from different experiments, we can add one further datapoint about inter-experimental consistency from a publication that has become available since our initial submission. Moris et al. (2024, BMC Biology) measured desiccation-induced differential expression in the same clone of *A. vaga* used by us and by Hecox-Lea and Mark Welch (2018). Their animals were desiccated over 37 hours, kept dry for 14 days then had RNA extracted 1.5h after rehydration to identify genes that were upregulated. They report a significant enrichment for HGTc among genes specifically upregulated during recovery from desiccation stress. The frequency of HGTc they report is 14.3% (37/258). This matches almost exactly with our results from reanalysing the 2018 desiccation experiment: 14.7% (266/1807). The OR we calculate for HGTc enrichment in the 2018 experiment is 1.3 (95% CI 1.1-1.5); for the study of Moris et al. (with a lower baseline HGTc value), the OR would be 1.58 (1.1-2.3), again representing a close overlap. A key result about HGTc proportions therefore appears robust to the following differences between these experiments:

- Conducted in different laboratories using lines evolving separately for at least 5 years.
- Entry into complete desiccation lasted 10 days versus 37 hours.
- Duration of desiccation was 7 versus 14 days.

- Time since rehydration was 60 minutes versus 90 minutes.
- Different thresholds were used to define differential expression
- Different methods were used to delineate HGTc genes

Moris et al. also measured HGTc enrichment among genes upregulated in response to ionizing radiation, another abiotic stressor. There was no significant enrichment for HGTc (67/648, 10.3%) in the radiation set, even though most of the radiation treatments also included a desiccation step. We take this as further evidence that enrichment of HGTc does not necessarily generalise to other stressors and is not simply a correlate of increasing levels of stress. We have added a brief discussion of this line of evidence in the text under the heading 'Enrichment of non-metazoan genes is stronger when responding to pathogens than to desiccation' (lines 311-317).

3. The criteria for choosing the post-infection time points are spelled out clearly in the Supplementary Information (lines 186-190), but I didn't see them in the main text. This information helps readers understand the rationale for these experiments, so it seems that the audience would be best served by including these criteria in the main text.

We have rewritten the first two paragraphs of the Results to explain the rationale for choosing these timepoints, with reference to the Supplementary Information mentioned by the Reviewer, and to the classic literature describing *Rotiferophthora* infection, and also to the new Supplementary Fig. 1 where we show visual examples of the progress of a successful fungal infection at these two timepoints.

4. Similarly, many readers will immediately note that the horizontally transferred genes are proportionately up- and down-regulated (i.e., there is no stronger enrichment of upregulated vs. downregulated genes). The authors provide thoughtful discussion of why this might be the case in a Supplementary Note. Not every reader will take the time to go through a 45-page supplement, so it seems a shame not to include these thoughts in the main Discussion instead.

We are glad that Reviewer draws attention to our discussion of this question. This had been put in the supplement to keep the manuscript concise, but we agree it comments on an important observation and so have restored a trimmed version to the main Discussion.

Reviewer #2:

This study meticulously delves into the presence/roles of horizontally transferred genes (HGT) in the coevolutionary defense mechanisms of bdelloid rotifers against their pathogenic adversaries, particularly the fungus *R. globospora*. Using a combination of infection assays, novel and pre-existing transcriptomic datasets, and comprehensive bioinformatics techniques (functional prediction tools; phylogeny; ...), the authors offer valuable insights into the puzzling evolutionary persistence and significant fraction (+-10%) of HGT within bdelloid genomes.

We thank Reviewer #2 for this positive characterisation of our study, and of our efforts to apply a range of techniques to these evolutionary puzzles.

Past studies have highlighted the prevalence of HGT in bdelloid species. However, the processes by which these genes were acquired and the evolutionary rationale for their retention in the genome are not entirely clear. Several hypotheses have been advanced, but empirical, particularly functional, data to support these theories is limited.

In this paper, two related species (and morphologically “very similar”), *Adineta vaga* and *A. ricciae*, were chosen to investigate transcriptional changes following fungal exposure. It was observed that *A. ricciae* demonstrated a higher resilience to the fungal infection when compared to *A. vaga*. To explore this further, transcriptional responses at 7 hours and 24 hours post-infection were examined. These time points consistent with microscopic and infection assay data, indicated changes in the transcriptomes of both species. A significant finding was the enrichment of HGT in differentially expressed genes, which might be involved in the rotifers' response to infections. Additionally, when this data was compared with prior transcriptomic studies on *A. vaga*'s desiccation and rehydration, the study highlighted the distinct roles HGTs might play in different physiological scenarios. Then, a Gene Ontology analysis of differentially expressed HGTs suggested specific patterns in *A. ricciae* that could be linked to its fungal resistance. Further examination identified a higher occurrence of NRP/PKS proteins in *A. ricciae* than in *A. vaga*. Predictive tools used in the study suggest that these genes might have a role in producing compounds similar to known antibiotics. RNA ligases and glucanases were also identified in *A. ricciae* as candidate genes for pathogen resistance. The paper concludes with the hypothesis that the HGT associated with pathogen defense might have been inherited from a common bdelloid ancestor, a suggestion supported by orthologous studies across various bdelloid species.

The manuscript is structured with commendable coherence, demonstrating a logical progression that effectively anticipates and addresses the reader's queries. Any questions arising at the end of a section are adeptly answered in the subsequent one. The meticulous details in the Methods & Materials, along with the supplementary data, underscore the authors' rigorous approach throughout the experimentation process. They have evidently taken measures to account for potential biases. This thoroughness not only underscores the authors' professionalism but also instills confidence in the presented results.

We thank Reviewer #2 for this complimentary assessment, and are encouraged that the time spent carefully checking the robustness of our results to a range of alternative parameters and interpretations has been appreciated.

While earlier research primarily highlighted the substantial presence of HGT in bdelloids or focused on a select few genes possibly acquired via HGT, this study stands out as the first to offer compelling evidence that HGT plays a pivotal role in the evolution and sustenance of asexual bdelloids, especially in their response to pathogens. Although the study lacks genetic tools for functional

validation – a limitation candidly acknowledged by the authors – the robustness of the data presented substantiates the core thesis put forth by the authors, within the confines of the current knowledge and tools available for bdelloid rotifers.

We are grateful to the Reviewer for highlighting the importance of our main findings, especially in the context of the evolutionary possibilities of this unusual study system, and its practical limitations.

Below are some comments that could enhance the manuscript. From my perspective, these improvements can make the content more accessible to readers unfamiliar with bdelloids, streamlining comprehension without compromising the quality of the work.

I would suggest the authors to not hesitate to provide some extra definitions in their manuscript to clarify the context of the paper and ensure easy access to scientific audience. For example, line 75 authors refer to parthenogenetic eggs. A short definition of this association may be relevant.

See below.

On Line 81: Consider including the recent publication by Terwagne et al. (2022) that explores the occurrence of a modified meiotic process in bdelloid germ cells. This work suggests that reproduction isn't strictly clonal in these organisms: Terwagne, M., Nicolas, E., Hespels, B., Herter, L., Virgo, J., Demazy, C., Heuskin, A.C., Hallet, B., & Van Doninck, K. (2022). DNA repair during nonreductional meiosis in the asexual rotifer *Adineta vaga*. *Science Advances*, 8(48), eadc8829. doi: 10.1126/sciadv.adc8829.

We now cite this paper and specify that the parthenogenetic eggs are produced by “an abortive, nonreductional meiosis” (at least in *A. vaga*). Together with the discussion of the absence of males and sperm, we hope this will be clear to readers. We hesitate to add more detail about mechanisms of oogenesis and reproduction because our work is mainly concerned with horizontal gene transfer.

On Line 97: For a more neutral presentation of your hypothesis, consider omitting the word “particularly”. While the introduction does provide evidence suggesting a role of HGT in the bdelloid immune system, prior to this study, there was no basis to anticipate whether the bdelloid genome might have an enriched fraction of HGT or merely a handful of such genes.

The word “particularly” has been deleted.

On Line 106: The use of the term “disproportionately” seems repetitive and might come across as overly emphatic in various sections of the manuscript. While it's clear from the paper that there's a notable enrichment of HGT in the described immune response, a more neutral phrasing might be more appropriate. Given that we are concluding the introduction section, consider referencing a "potential enrichment" or "increased likelihood" instead or better alternative to find by the authors.

We have changed this instance of the term as suggested, to “potential enrichment”. We reviewed uses of “disproportionate*” elsewhere, and deleted instances that were previously on lines 106, 122, 313, 379, 521 and 594.

On line 122: see my reference line 106.

This has been rephrased to the more neutral “differently”.

On line 134: I was unable to find how author were able to conclude that >95% of animals ingested spores within minutes of exposure. Please clarify in the M&M section how it was reported.

These details are now clarified in two places in the Supplementary Methods. First, the section “RNA-seq experimental design” has been updated to explain that we examined a small subsample of rotifers from each of the 32 extraction tubes between 40 and 60 minutes after inoculation (total $n = 484$). We found >95% contracted animals in experimental tubes and almost none in the UV-inactivated controls. We have updated the main text to be more specific about the 60-minute timeframe. Second, the section “Infection assays” has been updated with evidence to support the inference that contracted animals have ingested fungal spores. Specifically, 100% of contracted animals that we sacrificed and examined microscopically between 4 and 7h after inoculation had ingested at least one fungal spore.

On Line 138: The origins of *A. vaga* from moss in Italy and *A. ricciae* from a billabong in Australia, as detailed in the supplementary methods, suggest distinct ecological environments for the two species. Labelling these as similar might be misleading. The moss habitat of *A. vaga* can be characterized as limno-terrestrial, whereas *A. ricciae* was found in a pond environment, implying notable ecological differences. I'd recommend omitting this similarity claim or providing additional reasoning to substantiate the comparison.

As recommended, we have omitted this claim of ecological similarity.

Furthermore, this line prompts me to question the choice of models used. Why were these two related species specifically selected?

In the Supplementary Methods section ‘Rotifer and pathogen isolates’, we now explain that these species were selected because they were the only two rotifer lineages in laboratory culture for which assembled genomes were available at the time of the experiments (2019). The genus *Adineta* was known to be susceptible to the fungal pathogen *R. globospora* based on prior publications. Both rotifers are established in the literature as laboratory and genomic models.

Can we deduce the timeline of their divergence from common ancestors?

Estimating divergence times is an imprecise exercise in a group with almost no fossil record, and the present manuscript does not attempt this. It would not affect the conclusions, which are framed in relative rather than absolute timescales. In the section “Horizontally acquired pathogen-induced genes were largely inherited from a common ancestor”, we note that acquisitions common to different bdelloid genera must have occurred “millions of years ago”, based on an approximate molecular clock in the cited reference (Tang et al. 2014, Evolution). Based on more recent data, Nowell et al. 2021 (eLife, Fig. 2 Supplement 3) depict similar crown ages for the genera *Adineta* and *Rotaria*. Cross-referencing with Eyres et al. 2015 (BMC Biol., Fig. 2), an approximate maximum divergence time for *A. ricciae* and *A. vaga* would be 25 Ma, with a lower estimate closer to 10 Ma, but we would have very low confidence in these values, and the manuscript makes no references to specific timelines.

Incorporating another bdelloid species might offer broader insights into both transcriptional responses and pathogen resistance.

We agree and have since conducted further experiments of this nature with other bdelloid species. However, the present manuscript reports results from a particular set of experiments conducted contemporaneously with these two species, with care taken to ensure other factors were equal. We think these experiments provide sufficient evidence for our main conclusions and comparisons. While we are keen to share our findings from other species in due course, we are mindful of the length and complexity of the current manuscript, and of Reviewer #1's remarks about caveats and

considerations when comparing different experiments. In our view, it would be challenging to append comparative analyses of a third species while maintaining the readability and rigor that Reviewer #2 commends above.

Exploring these responses in a more expansive context could either corroborate or challenge the findings of the current study. It might be valuable to incorporate these considerations either in the discussion or supplementary methodology sections.

As recommended, we now mention the future value of expanding to other species in two places in the Discussion (lines 706 and 711-713). Preliminary results from other bdelloid species appear to corroborate the findings of the current study, but for reasons outlined above, we do not think these could readily be attached to the current manuscript, which describes a set of comparisons based on contemporaneous experiments using the two best-known bdelloid genomic models, supplemented with references to parallel experiments on the same species and lineages by other groups.

On Line 168: The authors centre their transcriptomic analysis on bdelloid HGT. Yet, even with the raw data accessible, there is a notable absence of information concerning the non-HGT response. While I grasp the authors' intent to concentrate on HGT, offering a concise overview of these results in the supplementary information could be beneficial for two main reasons: 1) to comprehensively document the immune response of bdelloids to fungi, and 2) to draw comparisons with the response specific to HGT. This point is addressed at the end of the manuscript. If there's no anticipation of featuring these details in a subsequent publication, I recommend (but not mandatory) that the authors incorporate (even a brief overview) them here for a more comprehensive understanding.

Reviewer #2 rightly notes that this manuscript centres on HGT components of the pathogen response. This reflects: 1) the motivating question, hypotheses and predictions established in the Introduction; 2) our attempts to relate these to what is (arguably) the most remarkable feature of bdelloid rotifer genomes; 3) the fact that the NRP/PKS clusters emerged consistently and spontaneously at the top of lists of overrepresented genes in the resistant species, regardless of whether we filtered for HGTc or not (lines 349-352; Supplementary Methods lines 417-428). In this sense, the rotifers themselves indicated what to concentrate on.

We agree that an important future goal will be to fully characterise the immune response of bdelloids (including the non-HGT components, which still form a large majority of the differentially expressed transcripts). However, we think it would be quite difficult to expand the present manuscript “to comprehensively document the immune response of bdelloids” and then to digest this down to a “brief overview”, especially given how little is currently known of this topic and how far this would extend beyond the context we have presented. Instead, we do indeed plan a subsequent publication focused on the broader transcriptional response.

For now, we have extended the accessibility of the raw data as the Reviewer suggests, to address the absence of information about the non-HGT response. Specifically, Supplementary Data 6 now lists enriched functions for all genes, including those of shared metazoan origin.

On Line 171: Can you confirm here that control populations were in contact with inactivated fungi?

We can confirm that the control populations were exposed to identical aliquots of fungal spores that had been inactivated by UV-irradiation. This is specified in the caption of Fig. 1, the main text Methods (section “RNA-seq experimental design”) and also in the Supplementary Methods.

On Line 186: Check Suppl Fig 2. Please italicized species name. Do for whole document.

This has been done.

One Line 185: I would remove the part “as expected if defensive or recognition pathways are conserved” as we have here no evidence at this stage that the differentially expressed genes are here involved in defensive or recognition pathways. Please adapt end of the section accordingly.

We have edited this line to read “as expected if pathogen response pathways are conserved.” This removes the unevicenced implication that shared genes responding to pathogen exposure are involved specifically in defensive or recognition pathways.

On Line 193: As outlined in the Methods & Materials section, the authors diligently validated their HGT using a recent and stringent methodology. However, when juxtaposed with previously published data, I noticed that the *A. vaga* genome in Simion et al., 2021 indicated an approximate 8.3% HGT. While the disparity between the two studies isn't substantial, I'm curious: did the authors briefly investigate the potential impact of alternative annotations on their results?

We agree that the value of HGTc = 8.3% from Simion et al. and our value (11.5%) is not a substantial difference, and is most likely driven by a combination of small differences in methodologies and threshold parameters between the two approaches, and the quality of the genome assemblies being assessed: Av20, analysed by Simion et al., is a near-fully chromosomal assembly, whereas Av13, analysed here, is more fragmented. While increased fragmentation may inflate the number of predicted genes in a genome (and by extension, the proportion of HGTc), in fact our initial investigations revealed a number of cases in our analyses where genes that looked like good HGT candidates had been falsely left off the HGTc list because of the overly-conservative criterion to require co-assembly of the HGTc with a gene of ‘unambiguous’ metazoan origin (i.e., that they are assembled on the same contig). This filter makes good sense to remove potential contaminants, but results in false-negatives when assembly fragmentation is high such that bona fide HGTc may be found on short scaffolds with no other genes. This is especially relevant when considering that many HGTc are in the subtelomeric regions of the genome, where assembly is already difficult. We discovered this issue during earlier versions of our analysis, and thus decided to modify the pipeline of Jaron et al. to cross-check all initial HGTc (based on BLAST similarities to non-metazoan homologs) against the assembled chromosomes of the Av20 genome that had since become available, retaining those which are encoded on bdelloid chromosomes (i.e., are clearly not contaminants). These details have been added to the Supplementary Methods (lines 368-377). Thus, the Av20 genome was critical in formulating final lists of HGTc for downstream analyses. While we did not directly assess the potential impact of alternative methodologies for classifying HGTc, we note the publication of a recent paper by Moris et al. (BMC Biol., 2024), which used the HGT pipeline of Simion et al. (‘Alienomics’) and came to almost identical conclusions about the proportion of HGTc upregulated versus desiccation in their results (~14.5%), as discussed in response to comment 2 of Reviewer #1.

On Line 200: A central argument in the paper, suggesting that HGT plays a role in the immune response of hydrated bdelloids when exposed to fungi, is the significant portion of post-infection overexpressed genes annotated as HGT. Nonetheless, it's noteworthy that HGT enrichment was also observed in downregulated genes across both time points and species.

Reviewers #1 and #2 agree on this point, so we have integrated the material discussing HGT enrichment for downregulated genes into the main text on lines 656-681.

Additionally, as highlighted later in the study, a considerable proportion of HGT is specific to the rehydration process in *A. vaga*. This context indicates that while many HGTs seem to be involved in the bdelloids' immune response, others (also many genes) may be linked with desiccation resistance

or other yet-to-be-determined functions. It would be beneficial if the authors could also reference, discuss and develop this in their manuscript, encouraging further investigation into the multifaceted roles of HGT in bdelloid rotifers.

We agree it is interesting that genes differentially expressed in the desiccation response also include a substantial proportion of HGTc (14.7%). This is a significant elevation compared with the baseline (OR 1.3, 95% CI: 1.1-1.5). It features in our manuscript primarily as a point of statistical comparison between biotic and abiotic stressors. We find that the HGT enrichment for the pathogen response is about twofold stronger than for desiccation across both species and timepoints (e.g. 24h *A. vaga* OR: 2.6, 95% CI:2.3-3.0). We do not discuss the identity or functions of the desiccation-specific genes further in this manuscript, but we reference the original 2018 paper from which that dataset is drawn, which discusses several examples in detail. We have now added references to recent and detailed studies by Nicolas et al. 2023 and Moris et al. 2024, which further highlight the role of HGT (and non-HGT) genes in resistance to desiccation and irradiation. Because the desiccation response is already being investigated and reported very ably elsewhere, we focus the present manuscript on primary evidence for the newly uncovered role of HGT in pathogen resistance and its implications. We hope both lines of work will contribute to further investigations and appreciation of the multifaceted role of HGT in bdelloid rotifers.

On Line 255: “..” check typo error.

Fixed.

On Line 278, Fig.3.a: In the legend, I would recommend including a full description of terms such as “NS core” for clarity. Throughout various sections of the manuscript, it occasionally becomes challenging to trace the origin or context of specific gene numbers. For instance, if “NS core” refers to non-differentially expressed genes, there seems to be a discrepancy; the figure lists 57,379 genes under “NS core”, while another table in the manuscript references a gene set of 35,079.

We have added a full description of ‘NS core’, ‘DE Core’ and other terms to the caption of Fig. 2, where these terms first appear, as quoted here:

“Classifications shown in legends are defined as follows: ‘NS core’ = no significant change in gene expression and is not HGTc; ‘NS HGTc’ = no significant change in gene expression and is HGTc; ‘DE core’ = significant change in gene expression (either up or down) and is not HGTc; ‘DE HGTc’ = significant change in gene expression (either up or down) and is HGTc; values indicate the number of genes falling into each category.”

For the sake of brevity, we decided not to repeat these definitions in the Fig. 3 caption.

We think the apparent discrepancy in gene number (e.g. ‘NS core’ in Fig. 3a = 57,379 vs 35,079 in Supplementary Table 10) is a misunderstanding. The value of 35,079 pertains to the number of predicted genes arising from a de novo transcriptome assembly of the RNA-seq data; these ‘genes’ were only used as a basic QC check on the RNA-seq dataset itself, not in subsequent downstream analysis. This is made explicit on line 304 of the Supplementary Methods to avoid potential confusion. All DE analyses presented in this manuscript use the gene models predicted from the two genomes of *A. vaga* (Flot et al., 2013) and *A. ricciae* (Nowell et al., 2018), so the total number of genes is 66,273 and 58,423, respectively (after some basic filtering for gene length). This is the sum of the 4 categories in the figure legends of the volcano plots shown in Figs. 2 and 3. We hope these totals are now communicated more clearly.

Furthermore, part c, despite the evident effort, is somewhat complex to decipher. The presentation of data could benefit from additional clarity. Mixing raw numbers with percentages can be particularly confusing for readers, so a more streamlined presentation would enhance comprehension.

Part c of Fig. 3 has been replaced with more intuitive Venn diagrams showing the extent of gene sharing among the different treatment groups across the two experiments. The original table has been retained in the Supplementary Information (Supplementary Table 5), since there is information here that is not conveyed in the Venn diagrams which may be useful to some readers.

On Line 378: While certain details are furnished in the Methods & Materials section, the current presentation leaves ambiguities about how the data can be employed to discern the mentioned "40 clusters." My interpretation of Figure 5c does not align with this representation, and further clarity is needed for a holistic understanding of Supplementary Data 8.

The Reviewer has astutely spotted two errors in Fig. 5c and Supplementary Data 8 (now 9). First, Fig. 5c was missing Cluster 3.8, which was present in the datasheet, and is now plotted correctly. Second, we now recognise that the former "cluster 6.6" is evidently composed of two separate clusters, nearby but in opposite orientations; these entities have now been correctly separated in both the figure and the data, as 6.6 and 6.7. Fig. 5c has been re-plotted and now shows all 40 predicted clusters discussed in the text, in addition to correcting one subtelomeric boundary. Other values and calculations have been adjusted accordingly. Discrepancies from the earlier version are minor and do not affect our focal upregulated annotations or other results.

We have added a new sheet to Supplementary Data 9 with further clarifications of terms and abbreviations in each sheet, to help readers link the annotated clusters to the Methods steps described on lines 452-504 of the Supplement. As we write on line 491 of the Supplement and line 602 of the main text, even in a highly contiguous assembly there are challenges in delineating and annotating these large and complex clusters, especially without transcriptional guidance since rather few are constitutively expressed. As we point out in the new sheet in Supplementary Data 9, several genomic clusters appear to be 3' incomplete at the ends of chromosomal contigs, indicating a need for further work to clarify clusters encoded on subtelomeric contigs that were not part of the haploid assembly. The particular difficulties posed in assembling and delineating clusters in subtelomeric regions are discussed on line 702. We have tried throughout to avoid language that might seem to claim a definitive delineation of all clusters (e.g. "approximately 40 clusters"). We hope these updates give a more holistic understanding of how the clusters were delineated.

I'd also like to use this comment to suggest that the authors furnish more comprehensive context with their supplementary data. In its current state, readers often encounter excel sheets that can be challenging to decipher due to the absence of adequate descriptions. Offering a clearer breakdown or accompanying explanations would significantly aid in interpretation and understanding.

We have added "README" cover pages to all Supplementary Data items with descriptions of what is included in the item and definitions of any terms used (column headings etc). An exception is Supplementary Data 10 "SeMPI 2.0 metabolite prediction" which is a zip archive of standard output (described here: <https://sempi-2-docu.readthedocs.io/en/latest/>). We hope this aids interpretation of these supplementary items.

Maybe I'm wrong but I think that complete differentially analysis of gene expression (excel sheet or DESeq2 output) for fungi response and desiccation is lacking (data are only available for HGT or restricted amount of genes).

We have included full results for gene expression analysis (raw counts, normalized gene expression data and collated results showing log₂ fold-change, FDR etc) as Supplementary Data 1.

On Line 443: In the manuscript, the authors emphasize the heightened resistance of *A. ricciae* compared to *A. vaga* and also demonstrate the associated transcriptomic response. However, in this section, where there's an attempt to functionally characterize candidate genes, the reference is predominantly to *A. vaga*. What prompted the decision to base the analysis on *A. vaga* rather than on the more resistant *A. ricciae*? It would be beneficial to readers if the rationale behind this choice is elucidated in the manuscript.

We used *A. vaga* for this analysis because the focal biosynthetic cluster in Av13 was fully assembled as the upregulated annotation “AVAG|g48151”, whereas the Ar18 orthologs (even though more numerous and more strongly upregulated) are assembled in fragments and/or as separate annotations, as shown in in Fig. 5b. As discussed in the Supplementary Material (“Orthology and copy number estimate for the upregulated NRP/PKS cluster”), and associated Supplementary Table 7, we were able to link the *A. ricciae* annotations and estimate the number of full-length copies by studying unique flanking sequences. However, the lack of a full-length annotation encoding the orthologous PKS-NRPS in *A. ricciae* meant that we turned to *A. vaga* for the SEMPI analysis of this cluster. We now elucidate this rationale as suggested: in the manuscript main text, in the caption to Figure 5 and in the Supplementary Material.

On Line 492: This section could benefit from further clarification to strengthen the assertions made. For instance, the reference to “other sequenced bdelloid genomes” remains vague (only short name/ not complete or codes available in suppl data). It would be helpful to the readers if more details about these genomes could be provided, preferably within the supplementary data descriptions. For example, in its current state, Supplementary Data 4 might suggest the ancient acquisition of HGT; however, without more explicit information in the manuscript or supplementary data descriptions, drawing the authors' conclusions becomes challenging. I recommend elaborating on this.

We have added a README page to Supplementary Data 12 (previously 4) which provides further information and context for these results, including a table linking the short codes with full species names and GenBank accessions to the published data. A new section in the Supplementary Methods describes how these results were produced (lines 381-387). Briefly, we used a simple Diamond ‘blastp’ approach to test explicitly for presence/absence of each significantly upregulated HGTc gene (for both species, at both timepoints), searching against the combined predicted proteomes of all bdelloid samples with available data. The number of proteins showing a significant hit in a given proteome is taken as the number of orthologs present in that sample, for a given HGTc query (tabulated in tabs 3-6 of Supplementary Data 6). We hope these additional descriptions clear up confusion regarding this part of the analysis.

While checking the manuscript, I noted a recurring theme which could lead to some confusion: genes identified in *A. ricciae*, presumed to play a pivotal role in its resistance against the fungi, are also observed in *A. vaga* and other bdelloid species. This raises several questions: Why don't these genes confer similar resistance in *A. vaga* (or were not enriched)? Why maintain these HGT in *A. vaga* if they don't provide adequate protection against the fungi?

Although we focus on the marked difference in resistance between *A. vaga* and *A. ricciae*, Fig. 1b still shows that approximately 30% of *A. vaga* have continued to resist the fungal infection for at least 72 hours. Whether this provides adequate protection would depend on the ecological and selective context, which we do not examine here. As we remark on lines 91-96 of the Supplement, “Even

partial resistance at an early stage could dramatically slow the spread of an epidemic, giving clonal relatives time to escape the infested habitat before the local population is exterminated,” and we cite a study illustrating this dynamic in *A. vaga*. The genes encoded and upregulated in both species may still be maintained in nature because they participate in the response to pathogens, even if the outcome in this case differs quantitatively between the two species.

As to why resistances might differ when both species encode orthologous genes, we can take the focal PKS-NRPS cluster as an example. Although both *A. vaga* and *A. ricciae* encode orthologs of this cluster, there are more genomic copies in *A. ricciae* and these copies are upregulated an order of magnitude more strongly. If this gene does play a pivotal role in resistance, then one hypothesis is that it confers less resistance in *A. vaga* because it is not upregulated as promptly or strongly in response to this pathogen. To generalise beyond this example, Figures 1c and 1d indicate that the pathogen response in *A. ricciae* is more rapid and extensive than that of *A. vaga* overall, whether considering total gene expression patterns or highly differentially expressed genes. These expression differences might help explain why resistance differences are seen between species sharing a complement of orthologous genes. We focus here on the biosynthetic genes because their expression differences were the starkest to emerge from our analyses.

Ultimately, the genetic basis of phenotypic resistance is likely to be more nuanced than assessing the presence or absence of a complement of genes, or how strongly such genes are expressed. It will be governed by coevolution between the host and pathogens, which may alter the expression, copy number or sequence of any of the thousands of genes involved in the interaction. On line 689 of the Discussion, we make a similar point to Reviewer #2: given the broadly shared complement of HGTc between species, gains and losses of HGTc are unlikely to be sufficient to drive coevolution with pathogens in the short term. At Reviewer #3’s invitation, we discuss how shorter-term ecological dynamics might help maintain the efficacy of genes encoding resistance functions, and buy the rotifers time to acquire new ones.

How do these genes function in other bdelloid species? Are there any indications that *A. vaga* might be more resilient than *A. ricciae* when faced with different pathogens? How do other species fare in similar scenarios? Addressing some of these points in the discussion section could enrich the narrative and insights of the article.

While we cannot speculate about gene function or resilience in different host and pathogen species that have not been tested here, we agree that these questions are critical to future empirical investigations of the coevolutionary process described above, now that the door has been opened by the present study. We have added this point to the Discussion as suggested, on lines 711-713.

On Line 521: I'd like to refer back to the earlier mention about the use of “disproportionate.” Considering that there is evidence that other HGTs might also be enriched in the rehydration response, it's crucial to tread carefully.

We have changed this to “markedly enriched in the response of bdelloid rotifers to pathogens”, following language recommended by the reviewer above.

The manuscript, in its current state, might inadvertently lead readers to believe that the primary driver for HGT acquisition in bdelloids is solely interspecific conflicts. Yet, a significant portion of HGT was not linked to pathogen responses. It might be enlightening at this juncture to present a more comprehensive overview of the data. For example: “Within the *A. ricciae* genome, X% of genes are identified as HGT. By T7, Y% of these genes exhibit overexpression. Of the differentially expressed

genes, Z% are HGTs, with W% of total HGTs likely playing a role in the response to fungi.” This broader perspective can offer a more nuanced understanding of the varied roles of HGTs in bdelloids.

In addition to changing the wording as suggested, we have taken the Reviewer’s advice and added these details to the manuscript. We feel it is too quantitative for the Discussion, but it can now be found in the Results at line 226:

“Of the total HGT_c complement in *A. ricciae* and *A. vaga*, 4.2% and 2.1% were differentially expressed at T7, rising to 9% and 5.7% respectively at T24.”

The other percentages requested can already be found in the same paragraph. In the caption to Fig. 2, where we discuss alternative DE fold-change thresholds, we also note that:

“At the 1.5-fold threshold, the proportion of all HGT_c that were differentially expressed in response to the pathogen at T24 was 19.7% for *A. ricciae* and 11.5% for *A. vaga*.”

We hope this helps avoid the impression that HGT acquisition by bdelloids is solely for the purpose of interspecific conflict, though it seems noteworthy that nearly 1 in 5 of the total complement of foreign genes encoded by *A. ricciae* seems to play some role in the response to the single pathogen thus far tested.

Lines 621-624: I would suggest the authors to develop a little this part that may be obvious for bdelloid experts but not for a wide audience.

We have rearranged and expanded this section to introduce and explain the ecological mechanism more clearly, and to incorporate Reviewer #3’s suggestion that we discuss links between spatiotemporal escape and the evolution of physiological immune mechanisms such as deployment of acquired antimicrobials. This material now comprises the penultimate paragraph of the Discussion and we hope it will be clear to a wider audience.

Supplementary Figure 11: Improve quality of figure.

We have replotted Supplementary Fig. 11 at higher resolution and improved the annotations and caption.

Reviewer #3:

The presence of HGT in bdelloid rotifers is nowadays well-established and this manuscript highlights some additional evidence of the importance of horizontally acquired genes in the evolutionary adaptation of bdelloid rotifers. Previous studies have shown the role of some non-metazoan genes in the radiation tolerance and epigenetic regulation of bdelloid rotifers. This article of Nowell et al. shows that some of their antimicrobial mechanisms over-represented during fungal attack are of bacterial type. This is important in the field of the evolution of sex and coevolutionary antagonism, and more broadly in the field of antimicrobial resistances.

We thank Reviewer #3 for this positive assessment of the importance of the study in various fields.

The transcriptomic analysis following pathogen exposure is solid: authors used three different packages to confirm the over-representation of HGTc in the DE genes and verified different fold changes (Supp Figs 3-8). They also compared with another study in which *A. vaga* was exposed to the abiotic stress “desiccation”, and where the enrichment of HGTc was lower. I however have a general comment on this comparison within *A. vaga* (see below).

The authors then identified the putative functions of some HGTc that are only upregulated in the bdelloid resistant species *A. ricciae*, with some unique bacterial polyketide and nonribosomal peptide synthetases that seem to produce antibiotics.

We are glad the Reviewer finds the transcriptomic analysis robust. We would just note that the more resistant species (*A. ricciae*, 80% survival) and the less resistant species (*A. vaga*, 30% survival) both upregulate orthologs of an HGTc cluster encoding the focal PKS-NRPS. The difference we report is that *A. ricciae* encodes and upregulates more copies, and it upregulates these far more strongly (Fig. 5b, Fig. 5e), so that the putative synthetase GO functions were only enriched in this species.

More general comments:

It would have been interesting to include a timepoint at 7 days following infection to have an idea of the reproductive capacity following survival. *A. ricciae* indeed had a higher survival rate than *A. vaga* at 72h following fungal exposure, and the authors mention in the Supp data “By 72h, the proportion of infected animals in 125 experimental wells appeared to have stabilised. In a subset of wells recounted at 96h, only a 126 small fraction of animals (~6.5%) had newly developed visible infections.” I wonder whether the *A. ricciae* population, triggering a strong resistant response at the start, i) was able to become pathogen-free after a few days or ii) did the population become weaker after several days and did the infection re-established? If the authors have some data on later timepoints, it would be interesting to include it in the supp data.

We have added Supplementary Fig. 16 to show the numbers of active, contracted and infected animals of each species in the subset of wells that were tracked up to 96 hours. This includes originally exposed animals that survived the fungus, as well as hatchlings arising from survivor reproduction as Reviewer #3 mentions. For control wells with UV-inactivated spores, the two species do not differ significantly, but for wells inoculated with live spores, *A. ricciae* had significantly more active individuals at T96 than *A. vaga*, by a factor of 10, as hatchlings began to emerge.

We do not have data for timepoints later than 96h because this study was designed to quantify physiological resistance of individual rotifers to an acute fungal challenge, not to track epidemic dynamics in populations over time. After 96h, surviving rotifers in the wells had begun to reproduce, as described on line 143 of the Supplement, so not all animals in the well had been exposed to the initial spore pulse under controlled conditions. After 96h, corpses infected by the fungus had begun

to deposit secondary pulses of spores into the wells, so spore densities were changing in nonlinear ways depending on the number of animals killed in the first round, and new exposures would depend on the ratio of these spores to new hatchlings (e.g. this was higher for *A. vaga*). By this point, we are no longer studying physiological resistance in the original animals but longer-term ecological dynamics, which have been reported in work published elsewhere (e.g. Wilson 2011 Biol J. Linn. Soc. for *A. vaga* versus *R. globospora*). Nevertheless, the tenfold difference in surviving animals and their offspring between *A. ricciae* and *A. vaga* in fungus-challenged wells at T96 suggests that longer-term dynamics would continue to strongly favour the more resistant species.

Even with longer tracking, the populations in our small assay wells could not have become entirely pathogen-free within 96 hours, or even 7 days, because the rotifers killed by the initial pulse of spores were still present, generating further spores to infect new hatchlings as described above (especially in *A. vaga*, where hatchlings appeared to ingest spores immediately and contract). In nature and in the lab (Wilson & Sherman 2010, Science; 2013, PRSL-B), populations become pathogen-free when animals desiccate and disperse in space or time away from infested patches. Infected corpses hosting the fungus are destroyed or left behind. For instance, Wilson (2011, Biol. J. Linn. Soc.) found that 66% of *A. vaga* populations that were dried for 28 days during an epidemic of *R. globospora* became pathogen-free. The probability of establishing a new, pathogen-free population increases if more animals survive the fungal attack for longer (Wilson & Sherman 2010). In this case, the number of surviving individuals of *A. ricciae* after 96 hours of exposure to the fungus is approximately tenfold higher than for *A. vaga*. Elsewhere, Reviewers #3 (and #2) requested further discussion of how resistance interacts with desiccation; this is now provided on lines 714-718, in addition to the material and citations mentioned above (Supplementary Methods, lines 94-96).

I am not so convinced by the difference observed in the transcriptomic response within *A. vaga* when comparing the biotic and abiotic stress. If you compare *A. vaga* biotic and abiotic response (Fig 4), is there a strong difference in functions between upregulated HGTc?

Yes, there is a strong difference in functions between upregulated HGTc. Fig. 4 is based on upregulated HGTc, and there is no overlap in the functional terms that are significantly enriched for the biotic and abiotic responses in *A. vaga*, except for the term “oxidoreductase activity”. We investigated this shared generic term more closely, and found that it disaggregates into three molecular function terms specific to the biotic stressor (GO:0016682; GO:0016705; GO:0016647), and one term specific to the abiotic stressor (GO:0016661), so again there is no overlap at this increased level of granularity (Supplementary Data 6). Even when we relaxed the statistical threshold for the desiccation dataset to $P < 0.1$, none of the molecular function terms we discuss in relation to biotic stress were found to be enriched in the abiotic desiccation response. In Figs. 3c and d, we also show that the differentially expressed HGTc genes themselves show little overlap between experiments (13%, Line 308).

In *A. vaga*, the upregulation of NRP/PKS functions was not detected,

This is not quite right, since upregulation of genes with putative NRP/PKS functions was detected in *A. vaga*. It might be helpful to distinguish two different sets of analyses and results: 1) upregulation of NRP/PKS genes; 2) functional enrichment of GO terms among those upregulated genes. In *A. vaga*, we detected significant upregulation of two NRP/PKS gene models at T24 in response to biotic stress (Fig. 5e), but this was insufficient to drive a significant functional enrichment for NRP/PKS terms in the GO analysis. In contrast, *A. ricciae* upregulated more NRP/PKS models and did so more strongly, resulting in highly significant enrichments for these biosynthetic functions in the GO analysis (Fig. 4). We later show that this corresponds to a genomic copy number expansion of the focal cluster.

... and while Table 1 shows a higher upregulation of HGTc when exposed to a biotic stress, the experiments were very different using *A. vaga* clones cultured in different laboratories and under different conditions.

In response to Comment 2 from Reviewer #1 above, we discuss three lines of evidence that evaluate the effect of differences between experiments, laboratories and conditions. Comparing the recent work of Moris et al. (2024, BMC Biol.) to our re-analysis of Hecox-Lea and Mark Welch (2018) is informative. Nearly identical HGTc proportions and relative odds of HGTc enrichment were seen in the response to desiccation in these two experiments, despite various differences between laboratories and conditions, as listed above. No significant enrichment for HGTc was detected in response to a second abiotic stressor: ionising radiation, when applied under several different conditions. To compare biotic and abiotic stressors more directly, we conducted a joint PCA of total gene expression for desiccation and pathogen attack (Supplementary Fig. 11), and found that both stress responses appear comparable in magnitude relative to their respective controls. Nevertheless, HGTc representation when exposed to the biotic stress is significantly higher than the desiccation response, as shown by comparing the odds ratios in Table 1 and their 95% confidence intervals. The odds ratios for the pathogen response are higher than those for desiccation by a factor of between 2 and 3. In recognition of potential uncertainty about the scale of this difference, we have adjusted the Abstract by deleting the word “far” from the line about stronger enrichment.

Therefore, can the authors highlight what are the main differences when comparing *A. vaga* exposed to biotic and abiotic stress, both for bdelloid-specific genes and for HGTc, including the putative functions identified? Some results are given for a few functions in the result section on RNA ligases and glucanases, a more detailed comparison is however required. This could be presented through a Venn diagram in Fig. 4 for example (comparing b, c, d).

The main differences in putative identified functions for HGTc genes between the biotic and abiotic stressors in *A. vaga* are discussed in the manuscript: in Figure 4; on lines 374-388; on line 543, on lines 429-448 in the Supplementary Information, in Supplementary Table 8, and can be checked in Supplementary Data 3 and Supplementary Data 5. We focus, as Reviewer #3 suggests, on the main differences, and have tried to avoid a known risk of functional enrichment analysis: a temptation to speculate post hoc about extensive lists of molecular functions with questionable relevance to the original predictions. This is important because there is almost no functional overlap between the biotic and abiotic sets, so we could choose to highlight or discuss any of approximately 59 terms that are enriched uniquely in the biotic response. These would be interesting targets for future investigation (and we do highlight selected genes in Supplementary Data 10), but here we focused on the prominent and a priori relevant differences seen for NRP/PKS functions, and spent our time further validating these differences with in-depth transcriptomic and comparative genomic analyses. We have not plotted a Venn diagram of overlapping enriched GO terms for HGTc genes because there are only three, but these are discussed in the Supplementary Methods on line 436.

As requested, we have performed the same functional analysis for all bdelloid genes, and the results are shown in a new Supplementary Data 6. Once again, there is almost no overlap between biotic and abiotic stress: only 6 terms out of 157 were shared. We refer to this result on lines 382-388. Finally, we examined the functional enrichment overlap between *A. vaga* entering and recovering from desiccation, and there were no overlapping terms to report or to plot at FDR < 0.05.

The title of the article should change, because this manuscript mainly highlights that more resistant bdelloid species deploy biosynthetic genes acquired from bacteria... than sensitive ones. It is not clear from the result section how similar is the functional response of *A. vaga* and *A. ricciae* to the fungal attack, and how different it is from the desiccation response within *A. vaga*.

We think this comment might arise from a misunderstanding that we addressed above. *A. ricciae* and *A. vaga* both resist the fungus, but *A. ricciae* does so much more strongly (Fig. 1b). *A. ricciae* and *A. vaga* both deploy biosynthetic genes acquired from bacteria when attacked by the fungus, but *A. ricciae* does so far more strongly (Fig. 5e). As discussed above, the Results section explains how the functional response to fungal attack is very different from the desiccation response within *A. vaga*. Specifically, there is almost no functional overlap between the two (Fig. 4, Supplementary Data 6).

We think the evidence we present supports the conclusion in the title that “Bdelloid rotifers deploy biosynthetic genes acquired from bacteria against attack by pathogenic fungi”. However, we have revisited the title as recommended and have simplified it to the following: “Bdelloid rotifers deploy horizontally acquired biosynthetic genes against a fungal pathogen”.

To identify whether the NRP/PKS genes found in *A. vaga* and *A. ricciae* are restricted to fungi and bacteria (Fig. 5), and which ones, or whether they are also found in other metazoans/eukaryotes (suggesting additional HGTc, see discussion lines 545-554), I advise the authors to provide the results of their BLAST analysis (BLASTp?) and to show the entire list of closely related sequences. The authors discuss this aspect in the discussion, without providing the entire list of blast results. The authors should also verify whether these genes are absent from monogonont rotifer genomes, a clade closely related to bdelloid rotifers, to confirm the acquisition within bdelloid rotifers specifically.

To double-check the closest matches to the putative NRP/PKS genes in bdelloids, we re-ran the BLAST analysis using the much larger UniProt/Uniref90 protein database (previous analyses used UniProt/Swiss-Prot). Full hit tables for this additional search are now provided in Supplementary Data 8, showing the top ~50 matches to the Uniref90 database for all 36 and 60 putative NRP/PKS sequences identified in *A. ricciae* and *A. vaga*, respectively. For both species, hits are overwhelmingly to bacterial proteins (~99% of total hits for either species; all top hits are to bacterial homologs). Of the few secondary hits to Eukaryota, the majority of these are to proteins from fungi or protists. There are no hits to metazoan sequences for *A. ricciae* or *A. vaga*, including to monogonont rotifers. Uniref90 top hits have been added to the condensation-domain phylogeny (Fig. 5a) for additional context to the diversity of putative NRP/PKS genes in bdelloids.

The absence of putative NRP/PKS genes in the monogonont rotifers *Brachionus plicatilis* and *B. calyciflorus*, and the acanthocephalan rotifer *Pomphorhynchus laevis* had been checked using the same survey strategy as for bdelloids (i.e., requiring HMMER hits to the three canonical domains AMP binding, PP binding and Condensation), but this had been omitted from the Supplementary Methods. No proteins from these other rotifer species pass this screen. Additional text has been added on lines 409-418 of the main text and lines 456-61 and 550 of the Supplementary Methods.

In this study, only two bdelloid species were studied, both being cultivated since decennia in laboratory conditions in the absence of pathogens. It would have been interesting to add a species sampled from the field, often exposed to pathogens. I wonder whether the expression profile is somehow biased, even in *A. ricciae*, because these species do not encounter pathogens since a long time. Nevertheless, the significant upregulation of HGTc when exposed to biotic stress in *A. ricciae* remains undeniable. Similar comment concerning a second pathogen. It would have been interesting

to test more than one pathogen and confirm this upregulation when exposed to different pathogen lineages, to show it is a more common response to biotic stresses.

As mentioned in response to Reviewer #2, we have since tested other pathogen and host species and early results are consistent with the findings described here, but we will describe that work in due course. We do not think it would be feasible to append a comparative analyses of multiple additional species to this manuscript while maintaining the readability and rigor that have been commended above, and we think the current evidence is sufficient to support our conclusions. We have added further remarks about the importance of testing multiple species on lines 711-713 of the Discussion. As Reviewer #3 observes, both *A. vaga* and *A. ricciae* have been in culture for 20-30 years, and it is interesting that both species retain and deploy biosynthetic gene clusters (and upregulate hundreds of other HGTc genes) against fungi, despite some time without exposure to selection from these pathogens. This might not be surprising if we presume that these parthenogenetically propagated animals are rather slow to evolve either gain or loss of functions, but such questions are outside the scope of the present study and would require different experiments.

On a more speculative note, how would other parthenogens like oribatid mites and tardigrades, also living in soil and semi-terrestrial environments, deal with pathogens while HGT is not prevalent in these asexual species? Would long periods of desiccation be sufficient, as shown in previous papers of Chris Wilson? Why do bdelloid rotifers need to co-opt additional pathways from bacteria? It would be interesting to discuss this comparison with other parthenogens thriving in similar habitats.

We agree with Reviewer #3 that it would be speculative to introduce comparative discussion of animal groups that are not studied here. However, we mention the springtail *Folsomia candida* as an example on lines 611-614. This also lives in soil and semi-terrestrial environments, and appears to have acquired one or two possible NRPS genes, which we now show in Fig. 5a. Although *F. candida* certainly can reproduce parthenogenetically, many populations are sexual and even the apparently asexual populations continue to produce males and sperm (e.g. Kampfraath et al. 2020, *Evol. Biol.*). The same holds for populations of putatively asexual oribatid mite species (e.g. *Platynocheilus peltifer*: Taberly 1988, *Acarologia*). So, one potential answer to the Reviewer's question is that these other organisms retain sufficient rates of sex to be able to coevolve with pathogens in the usual way via reshuffling of innate immune genes (lines 61-71), and therefore experience less pressure to acquire additional genes via horizontal transfer to deploy against pathogens. It is easy to think of several alternative hypotheses, but we think these speculations are beyond the scope of the current work, and perhaps better suited to a future review of the topic.

As requested (and also by Reviewer #2), we now discuss possible links to long periods of desiccation on lines 713-720. In that context, the comparison to semi-terrestrial tardigrades is indeed interesting, and this is already discussed in previous literature (Wilson & Sherman 2013, PRSL-B), cited here.

More specific comments:

- The paragraph (lines 50-53) on the antibiotic production in bacteria only focuses on NRPS and PKS, while it would be interesting here for the broader audience to give more examples.

We have cited a reference in this section and changed our wording to cover several other important examples of horizontally transferred antimicrobial machinery, including various mechanisms of toxin production, Type IV secretion systems and prophages (Granato et al., *Curr. Biol.* 2019). Most of these are not encountered in our work, so we give little further detail, and prefer to use the space to clarify the details of NRPS diversification requested below.

- This sentence (lines 54-55) is not clear to me (because I am not a specialist) "... their mobility and modular structure facilitates diversification to produce a vast array of secondary metabolites...". What type of mobility and modular structure, and which secondary metabolites are produced. Some clarification here on this interesting aspect should be given.

We have modified this section to clarify these points. We explain on line 53 that the set of secondary metabolites produced by NRP/PKS includes toxins and immunosuppressants as well as antimicrobial compounds (and many more). We have replaced 'modular structure' with a more accessible reference to the 'assembly line' organisation of NRPS modules. Mobility is discussed with reference to plasmids and lateral gene transfer within and between bacteria, which facilitates recombination between different modules, resulting in changes to the order and metabolite output of the 'assembly line'. We are now quicker to cite a reference to a comprehensive overview of this topic (line 56).

- The authors mention on line 61, "Although meiotic shuffling has different effects to HGT, ..." Which different effects? It is a different mechanism to generate genetic diversity, but it is not clear which effect the authors refer to.

We did not intend to refer to a single effect here, and there are many to consider. With HGT, "the DNA that is recombined by recipient cells consists of relatively short tracts, and not 50% of the genome as in meiotic sex" (Vos et al. 2019, Trends in Microbiology). Further, "unlike the sexual process of higher eukaryotes, bacterial sex does not require the formation or fusion of gametes, cell contact, recombination, or even the creation of genetic variation or new combinations of chromosomal genes, and it is not associated with reproduction. Sex in bacteria is simply the uptake of any genetic material that might eventually be vertically or horizontally transmitted" (Narra & Ochman 2006, Curr. Biol.). We allude to these differences when we write that in meiotic sex, "whole genomes are shuffled every generation though recombination, segregation and outcrossing." We have added a citation to support this point, but discussing further details of these population genetic and theoretical issues would be beyond the scope of the present work.

- Some seminal papers have recently been published on the reproductive mode of bdelloid rotifer *Adineta vaga*, showing a non-reductional modified meiosis I (Terwagne et al. Sci Adv) and suggesting recombination (Vakhrusheva et al. Nature Comm and Simion et al. Sci Adv). Recombination has important consequences, and I am surprised the authors do not mention those latest results here in the introduction (paragraph on lines 74-79).

We now cite Terwagne et al. and give further details of the non-reductional modified meiosis as the mechanism of parthenogenesis. We already cite the other two papers in this paragraph. We do not go into further detail because this manuscript is primarily about HGT and the physiological response to fungal pathogens, rather than recombination, chromosome pairing or population genetics.

- In the paragraph on HGT (lines 81-87) authors should highlight that even desiccation sensitive bdelloid species, of the genus *Rotaria*, also have a high number of HGTs in their genome.

We write that the same result is found "for all bdelloid genomes so far examined". This would include the genus *Rotaria*, and we feature six species of this genus in Supplementary Data 12. We are not sure why desiccation sensitivity would be relevant here, or what readers would gain from mentioning it, since this manuscript neither posits nor relies on any connection between desiccation and the acquisition of non-metazoan genes. We only use desiccation as one example of an abiotic stressor for statistical comparison to the pathogen response. Reviewer #2 encourages us to "make the content more accessible to readers unfamiliar with bdelloids", and we worry that introducing historic speculations about mechanisms of DNA incorporation would have the opposite effect.

- On lines 88-91, HGT is suggested to be too slow to equate sexual shuffling seen in typical eukaryotes, UNLESS horizontal exchanges do occur between bdelloid individuals. While this is still debated, horizontal exchanges during desiccation could still be an important mechanism of generating diversity in this animal clade.

This sentence and the paper it cites are discussing rates of horizontal gene acquisition from bacteria, fungi, plants and other sources. The present manuscript deals with these genes and their putative functions. It does not address or depend on a separate speculative hypothesis posed by Gladyshev et al. (2008, Science 320: 1210) about “horizontal exchanges during desiccation”. We know of no evidence that would support further comment on that interesting idea at present.

Materials and methods

- The infection assays were performed in sterilised distilled water WITHOUT food (line 649), such condition will induce the contraction of bdelloid individuals, even without pathogens. The authors mention in the results section (line 134) a rapid contraction of infected individuals, what about the negative control? Were there less contracted individuals when no pathogens were present in sterilised distilled water?

Yes, there were significantly fewer contracted individuals when no live pathogens were present in the sterilised distilled water. The relevant data are now shown in Supplementary Figure 16 for infection assay wells and discussed on lines 209-219 of the Supplementary Methods for inoculation tubes. Specifically, the mean rate of contraction in negative control wells was 5%, while in wells inoculated with live pathogens this was 90% for *A. ricciae* and 98% for *A. vaga* (no significant difference). In the inoculation tubes, we estimated that the proportion contracted with the live fungal inoculum was 97% for *A. ricciae* and 95.8% for *A. vaga* (no significant difference), versus <3% in negative control tubes. The animals in the negative control wells remained >90% active and <10% contracted throughout the experiment despite the absence of food. We think the rotifers might need to be deprived of food for longer than 4 days before contraction rates would substantially increase.

- In Supp Methods the authors mention (lines 86-88) “an individual host can resist the initial attack and prevent an ingested spore from establishing a successful infection, or at least delay its progression. Even partial resistance at an early stage could dramatically slow the spread of an epidemic...”. How is the resistance visible, the spore is ingested but never germinates? Or the spore is not ingested, or? It would be interesting the authors provide more information on the phenotypic resistance observed, especially in *A. ricciae*.

We now provide additional evidence and discussion of this point under ‘Infection assays’ in the Supplementary Methods, and in the new Supplementary Fig. 1. Contracted animals have ingested spores, and the spores are capable of germinating and forming assimilative hyphae in *A. ricciae* in at least some cases. The process is apparently interrupted in over 70% of animals at a point between spore ingestion and extensive colonisation of the animal by assimilative hyphae. Anecdotal evidence (Supplementary Fig. 1, lower panel) seems to suggest that even in those *A. ricciae* where assimilative hyphae are intruding, they are smaller in extent than for *A. vaga* at the same timepoint, which would be consistent with a physiological environment more hostile to fungal growth, but we lack the power and replication to evaluate this claim statistically. This will be a fruitful area for future work, because the present manuscript has focused on the transcriptomic and genetic rather than the cytological or microscopic aspects of the resistance response, which will require quite different techniques.

- Putative HGTc (line 718) could have been classified as in the paper Simion et al. Sci Adv using the Alienomics pipeline developed (see github). Alienomics considers the presence of other metazoan

genes on the scaffold, besides other important parameters of HGT, rather than one gene of unambiguous metazoan origin as in Jaron et al. 2021 (“we considered non-metazoan genes as HGT candidates only if they were on a scaffold that also encoded at least one gene of unambiguous metazoan origin, to control for potential contamination in the genome assemblies”)

We agree there are a number of good tools and pipelines now available for HGT detection, including Alienomics and AvP (Koutsovoulos et al. 2022, PLoS Comp. Biol. 18: e1010686), as well as the pipeline used by Jaron et al. 2021. These tools all use the same core principle of sequence similarity to ‘outgroup’ versus ‘ingroup’ proteins to infer candidate HGTs, but differ in which parameters and additional filters are applied to produce a final list of HGTc. We also agree that the criterion for a HGTc to be on a scaffold with at least one other gene of “unambiguous metazoan origin” may sometimes be inappropriate—in fact it can be overly conservative, especially when applied to draft genomes comprised of many scaffolds (as is the case here), because some genuine HGTc genes may be discarded if they occur on short scaffolds with no other genes present. This may be even more applicable for bdelloids because many HGTc are found in subtelomeric regions, which are difficult to assemble. To mitigate this potential bias, we did not include the criterion for colocation with at least one gene of “unambiguous metazoan origin” in our construction of an initial list of HGTc, but instead took the full list of initial HGTc (i.e., $h_u > 30$ and $CHS > 90\%$) from Jaron et al., and then retained only those genes which mapped to the recently available chromosomal-level Av20 assembly, to rule out potential contaminants in the Av13 and Ar18 assemblies (since contamination is extremely unlikely in the Av20 assembly). This is explained in the ‘HGT classification’ section of the Supplementary Methods, and we have added text on line 822 of the main text to draw attention to this detail.

Results

- Fig 1b. Points indicate replicate laboratory populations of *A. vaga* (red) and *A. ricciae* (blue); I count 20 blue points and 16 red points. In the M&M section it is mentioned “Adult rotifers were transferred to 96-well plates (Thermo-Fisher), with approximately 11 animals per well (mean: 11.0, SD: 3.5) in 60µL of sterilised, distilled water”. Can the authors be more precise here, even in the Supp methods it is not entirely clear. Were there 16 replicates for *A. vaga* and 20 for *A. ricciae*?

There were 16 replicates for *A. vaga* and 21 for *A. ricciae*, but one of the zero datapoints for *A. ricciae* was previously obscured by a failure of jittering that has now been corrected.

- How are the 216 *A. ricciae* and 189 *A. vaga* represented in Fig 1b?

The 216 *A. ricciae* represented in Fig. 1b were spread across 21 test wells, with approximately 10 animals per well exposed to a dose of the spore treatment. Each datapoint is the proportion of animals in a well that were killed by infection. We hope this is clear in the caption, which states:

“Points indicate replicate laboratory populations of *A. vaga* (red) and *A. ricciae* (blue) with approximately 10 individuals (range 6 to 20) exposed to 1000 conidia.”

- Within each well (represented by a point on Fig 1b?) there was no variability in mortality between individuals?

Within a typical well for *A. ricciae*, 7 animals survived and 3 animals were killed by infection at 72h. Such a well would then be plotted as a point at ‘0.7’ on the y-axis of Fig. 1b. We hope the new Supplementary Fig. 16 will help visualise this outcome.

- Is one point representing a population that increased in number (meaning it survived) or decreasing in number (meaning mortality)?

One point represents a population of approximately 10 individuals that was exposed to a dose of fungal pathogen spores and tracked for the following 72 hours. We tracked their fate for up to 72 hours because after this, some populations did indeed increase in number as new hatchlings emerged that had not been exposed to the spores. This is specified on lines 138-144 of the Supplementary Methods, and plotted in Supplementary Fig. 16, which also shows a subset of wells tracked up to 96 hours, including the arrival of unexposed hatchlings. No wells decreased in number, because we counted all rotifers regardless of whether they were active, contracted, infected or dead. Fungal mortality was measured as the proportion visibly killed by the fungus (lines 135-138).

- If there was variability (some individuals died, other reproduced within the same well), how is this represented in Fig. 1b within each point?

Within a typical well for *A. ricciae*, 7 animals survived and 3 animals were killed by infection at 72h. Such a well would then be plotted as a point at '0.7' on the y-axis of Fig. 1b. The 72h timescale of the experiment is too short to include reproduction (see Supplementary Fig. 16).

- Fig 1c. At T24h there was a significant difference between *A. vaga* and *A. ricciae*, only for the upregulated genes, this could be indicated with an * on the figure.

The error bars on Fig. 1c (now Fig. 1d) represent 95% CI from 1000 bootstrap replicate samples of the data (i.e. sampling genes with replacement and recalculating the number of genes falling out as statistically significantly DE based on the same thresholds). Thus, while the CIs provide a visual check on the robustness of the total number of DE up/down genes at each timepoint, they are not independent samples that could be used to test for statistically significant differences. However, we can instead use a simple Chi-square contingency test to ask whether the two species differ in the proportions of their total geneset that are significantly DE up or down at either timepoint, with a Bonferroni correction for four comparisons ($P < 0.0125$). Such a test shows that all the differences between species in Fig. 1d are highly significant, including for downregulated at T24h, and *A. ricciae* always has the stronger response, consistent with Fig. 1c. These results would support the asterisks Reviewer #3 requests, so we have added them and we explain this test briefly in the caption.

- Lines 180-188: what is the overlap in transcripts between T7 and T24? How many genes found at T7 are also detected at T24, a Venn diagram could represent this overlap. Supp Fig 1 shows the DE correlation.

The extent of overlap in gene identities between timepoints (within species) is now shown as Euler diagrams in Fig. 1 (part e), and between species (linking gene IDs through orthology) as Venn diagrams (Supplementary Fig. 3), and discussed on line 195-200 in the main text. These data show substantial sharing between timepoints and species, particularly for upregulated genes. Venn diagrams have also replaced the previous Fig. 3c, to show equivalent data specifically for *A. vaga* genes shared between the pathogen experiment and the desiccation experiment.

- Line 193: Jaron et al reported 10% for *A. vaga* and 10.6% for *A. ricciae*, Simion et al. reported 8.3% for *A. vaga*. What is this 11.5%?

As explained above, the 11.5% is derived from taking the 'long-list' of HGTC from Jaron et al., and retaining those HGTC which map to the chromosome-level Av20 assembly (see lines 368-377 of the Supplementary Methods). This was to mitigate the criterion of co-assembly of a HGTC with a gene of "unambiguous metazoan origin" (discussed above), which we found during preliminary investigations to be overly conservative (i.e., misclassifying some genes as non-HGTC when further analysis showed clear evidence for HGT).

- Fig. 4: it should be indicated in the title that Fig. 4a and b represent the timepoint 24h for both species. It would be clearer if the authors present the results of Fig. 4a and b in a Venn diagram to show what is common to both species in the upregulated HGTc response and what is specific to each species. Using colours and frames makes it less easy to interpret. Also compare the biotic and abiotic stress conditions only within *A. vaga* (see general comment above).

We have updated Fig. 4 to clarify that this reflects T24 for both species. As discussed in response to a comment above, there is effectively no overlap in enriched GO terms between the desiccation and pathogen responses within *A. vaga*, so a graphical comparison would not be informative. Given the small number of terms shown in Fig. 4a and b (11 and 6), we think the current presentation is more informative than a Venn diagram, because it also shows the names and proportional contributions of each category.

- Fig 5. a, the phylogeny could include more sequences of the blast analysis since NRP/PKS genes appear to be also known from a few animals (see discussion) and protists?

Additional sequences have been added to the phylogeny in Fig. 5a for further context, including the top-matching homologs resulting from the Diamond 'blastp' of putative NRP/PKS proteins vs Uniref90 (all bacterial; see reply to comment above), and selected rare cases of NRP/PKS found in animal genomes (e.g., Nrps-1 and Pks-1 in *Caenorhabditis elegans*, and recent examples found the springtail *Folsomia candida*, the mollusc *Lottia gigantea*, the lancelet *Brachiostoma belcheri*, and the sea star *Patiria miniata*). Details have been added to the caption of Fig. 5, on lines 409-416 of the main text, and on lines 540-552 of the Supplementary Methods.

REVIEWERS' COMMENTS

Reviewer #1 (Remarks to the Author):

The authors have very thoroughly addressed the concerns of all of the reviewers and the manuscript is suitable for publication.

Reviewer #3 (Remarks to the Author):

The authors have clearly addressed all the comments raised by the different reviewers and have improved the manuscript considerably. This revised manuscript can be accepted for publication.

RESPONSE TO REVIEWERS' COMMENTS

Reviewer #1 (Remarks to the Author):

The authors have very thoroughly addressed the concerns of all of the reviewers and the manuscript is suitable for publication.

Reviewer #3 (Remarks to the Author):

The authors have clearly addressed all the comments raised by the different reviewers and have improved the manuscript considerably. This revised manuscript can be accepted for publication.

RESPONSE:

We thank the Reviewers for their positive remarks on our revisions.